# Tropical cyclone intensification and extratropical transition under alternate climate conditions: a case study of Hurricane Ophelia (2017)

Marjolein Ribberink[1,2], Hylke de Vries[2], Nadia Bloemendaal[1,2], Michiel Baatsen[3], and Erik van Meijgaard[2]

[1]Institute for Environmental Studies, Vrije Universiteit Amsterdam, De Boelelaan 1111, 1081 HV Amsterdam
[2]Royal Netherlands Meteorological Institute, Utrechtseweg 297, 3731 GA De Bilt, the Netherlands
[3]Institute for Marine and Atmospheric Research, Utrecht University, Heidelberglaan 8, 3584 CS Utrecht, The Netherlands

**Correspondence:** Marjolein Ribberink (m.r.s.ribberink@vu.nl)

**Abstract.** Post-tropical cyclones can have substantial impacts on regions unaccustomed to such powerful storms. Previous studies have found that the prevalence of such storms is expected to increase in the future. Hurricane Ophelia in October 2017 presents a potential future analogue, reaching major hurricane strength far beyond current climatological boundaries and affecting Ireland as a powerful post-tropical cyclone. Here, we look at the changes in the structure, behavior, and impacts of Ophelia between current and possible future climate scenarios, with a focus on the extratropical transition phase. Using a regional model, we downscale GFS analysis data, and simulate alternate climatic conditions using a prescribed uniform temperature forcing. In warmer scenarios, Ophelia becomes a stronger and larger storm, while its path shifts westwards. These changes allow Ophelia to delay extratropical transition and maintain more of its tropical characteristics, increasing the impacts upon landfall. We also demonstrate that in the case of Ophelia, storm intensity and impact are very sensitive to initial conditions. Additionally, the tropical phase of the storm is more impacted by the change in temperature than the extratropical phase, indicating that proper tropical cyclone modelling is required for accurate predictions of post-tropical cyclones impacts. Our simulations indicate that rare cases similar to Ophelia likely present an even larger risk to affected areas in Western Europe in a warmer future.

## 1 Introduction

Post-tropical cyclones (PTCs) can pose a significant hazard when they make landfall in regions unaccustomed to these often-powerful storms. PTCs are a subset of extratropical cyclones (ETCs) and are the result of a tropical cyclone (TC) undergoing extratropical transition (ETT). ETT is the process by which TCs lose their tropical characteristics (symmetrical, vertically-stacked, warm convection-driven core), and adopt those of ETCs (asymmetrical, tilting with height, baroclinic instability-driven) as they move out of the tropics and interact with the midlatitude atmosphere, often by coming into contact with the subtropical jet stream (Hart and Evans, 2001; Jones et al., 2003). Approximately half of all TCs in the North Atlantic (NA) basin undergo ETT every year, and the resulting PTCs are responsible for a large amount of damage outside of the typical TC risk

zones (Bieli et al., 2019; Hart and Evans, 2001; Studholme et al., 2015). For example, Hurricane Sandy (2012) underwent ETT off the United States' Eastern Seaboard, during which it expanded greatly in size (Blake et al., 2013). When it subsequently made landfall in New Jersey it was more than 1600 km across and dropped rain and snow from North Carolina to Canada, as well as causing inundation across the length of the Eastern Seaboard. The highest flooding took place around the densely populated New York City metropolitan area, and in total the storm caused 88.5 billion USD (2024 adjusted) in damage in the United States alone (Blake et al., 2013; Smith, 2020).

Several recent studies showed that ETT will be affected by a changing climate (Bieli et al., 2020; Jung and Lackmann, 2019, 2021, 2023; Liu et al., 2017, 2020; Michaelis and Lackmann, 2019, 2021). Common changes to PTCs in simulations with warmer climates include lower central pressures, an increase in precipitation exceeding Clausius-Clapeyron expectations from just a warming atmosphere, and a poleward shift in track. Most of the studies on ETT with changing climate have focused on the impacts to the eastern United States, but several studies have shown that Western Europe is also projected to see an increase in TCs and PTCs as increasing sea surface temperatures (SSTs) expand the (P)TC genesis and occurrence regions (Baatsen et al., 2015; Dekker et al., 2018; Haarsma et al., 2013; Liu et al., 2017). Combined with the expected increase in storm strength, the expanding genesis regions heighten the chances that (P)TCs survive the journey to Europe. Additionally, PTCs that impact Europe are more likely to have undergone reintensification after their ETT compared to traditional ETCs, meaning that they are often stronger than their ETC counterparts (Sainsbury et al., 2022).

Baatsen et al. (2015), extending the research of Haarsma et al. (2013), analyzed TC-permitting high-resolution global climate simulations with applied RCP 4.5 climate conditions. They discovered that under warmer climatic conditions Western Europe will be impacted by more severe autumn storms, and this increase can be largely attributed to storms of tropical origin. They describe a hypothetical PTC that appeared in their simulations whose characteristics and lifecycle replicated many of the features seen in storms that make landfall in Europe. This storm, which they called Amy, is a TC that transitions to a warm-core PTC as it interacts with an upper-level trough in the midlatitudes. This causes Amy to reintensify as the trough steers the storm towards and over the British Isles.

Amy has a stunningly similar real-life equivalent in Hurricane Ophelia, which impacted the Azores and Ireland in October 2017 (Stewart, 2018). After undergoing extratropical transition under the influence of a midlatitude trough, now PTC Ophelia downed hundreds of trees and resulted in the death of three people. With its similarity to the Europe-impacting PTCs found by Baatsen et al. (2015), Ophelia is a representation of what we may expect more often with future warming. As such, being able to determine the changes this future warming may impart to such a storm is important to minimize future damages and loss of life.

There have been many case studies on PTCs in the last several decades, especially in the North Atlantic (Atallah and Bosart, 2003; Evans and Hart, 2008; Feser et al., 2015; Galarneau et al., 2013; Jung and Lackmann, 2019; McTaggart-Cowan et al., 2004; Thorncroft and Jones, 2000). However many of these focused on US-impacting storms, and those that do impact Europe are only studied in current climate conditions. As far as the authors are aware, no specific case studies have been done on Europe-impacting transitioning storms incorporating climate change factors. In this study, we therefore aim to fill this gap by examining the changes in the structure, behaviour, and impacts of Hurricane Ophelia under alternate climate

scenarios, paying particular attention to the changes in Ophelia's ETT. We run alternative climate scenarios at fixed degrees of temperature warming (the so-called ΔT approach) utilizing the Regional Atmosphere Community Model (RACMO) model. The results show that Ophelia becomes a larger and stronger storm under warmer climate conditions. The outcomes can be used to demonstrate the increased risk posed by the expected increase in frequency of such storms under climate change conditions.

Section 2 provides a description of the case study. The data and methods used in this paper are described in Section 3. We present the results of our analyses in Section 4 and discuss the study in Section 5. Finally we conclude the paper in Section 6

## 2 Case Study

Ophelia developed out of an upper-atmosphere trough that had been pushed south from the midlatitudes in early October 2017. A surface low established itself under the upper level divergence and induced upward motion east of the trough by 3 October 12 UTC (Stewart, 2018). Initially the low developed only shallow convection, but with SSTs under the area just higher than the development threshold (26°C), tropical development was able to occur (see Fig. 1(a)). The convection deepened, and by 11 October 18 UTC Ophelia had reached hurricane strength (10-minute sustained wind speeds of >110 km/h)(Simpson, 1974). An approaching mid-level trough picked up the slow-moving storm (see Fig. 1(b)) which was struggling with increasing wind shear and decreasing SSTs (Stewart, 2018). Embedding into the trough, Ophelia started restrengthening as it accelerated northeastward, peaking at an estimated minimum pressure of 957 hPa and maximum 10-minute sustained wind speeds of 172 km/h on 14 October. Ophelia reached Category 3 on the Saffir-Simpson Hurricane Wind Scale, making Ophelia the easternmost major hurricane to form in the North Atlantic since the satellite era (Simpson, 1974; Stewart, 2018).

As the jet accelerated it towards Europe and into colder waters, Ophelia started its ETT on the morning of 14 October. Passing by the Iberian Peninsula on 15 October, strong outflow winds from Ophelia fanned wildfires in Portugal, which claimed the lives of 51 people (San-Miguel-Ayanz et al., 2020). As the storm approached Ireland, it completed its ETT into a Shapiro-Keyser-type ETC, in which the warm front wraps around the cyclone centre while the cold front breaks off the main circulation (see Fig. 1(c)-(d); Shapiro and Keyser (1990)).

Ophelia made landfall on 16 October 2017 as a PTC, with 10-minute maximum sustained winds of 145 km/h, and brought the highest wind gust ever recorded in Ireland at 191 km/h (NDFEM, 2019). These high winds led to hundreds of downed trees, causing the storm's three fatalities and isolating outlying communities. The early fall timing of this storm meant more trees were in full foliage as opposed to the typical midlatitude winter storm, consequently the impact was likely greater than it would have been had it occurred later in the year (Moore, 2018). With hurricane season spanning June-November, this impact is one we can expect to see repeated with more Europe-impacting TCs and PTCs.

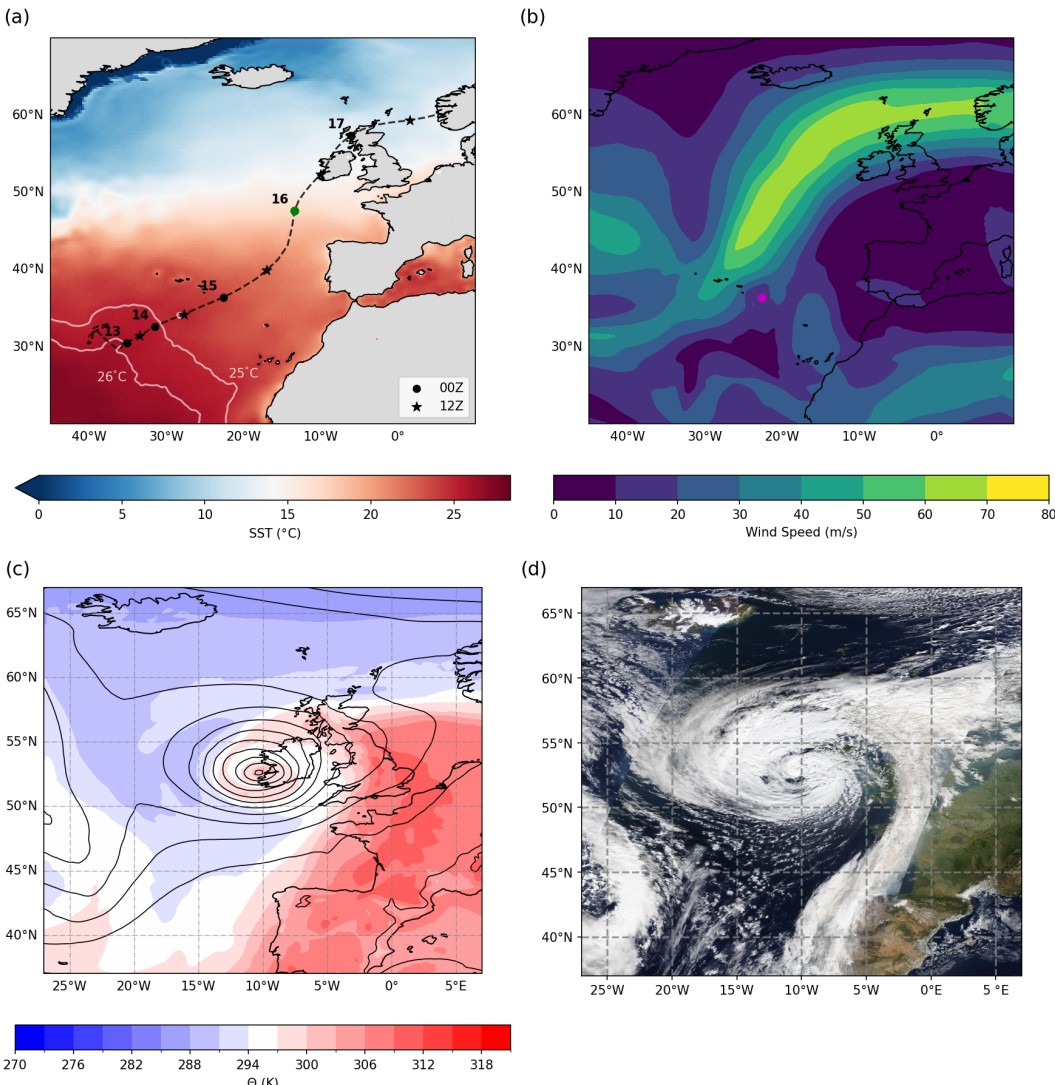

**Figure 1.** Composites of Hurricane Ophelia, (a), (b), (c) from GFS analysis, (d) from NASA Worldview. (a): National Hurricane Center best track with markers for the 00 UTC and 12 UTC positions of Ophelia indicated by dots and stars respectively, with day indicated at 00 UTC position, over sea surface temperature at 13 October 00 UTC with the 25 and 26°C isotherms contoured. Green marker is position at time of ETT completion. (b): 200 hPa wind speed at 15 October 00 UTC with the best track position of Ophelia indicated by the pink dot. (c): 850 hPa potential temperature (shading) with mean sea level pressure contours at 16 October 12 UTC. (d): Satellite image of Ophelia from approximately 16 October 12 UTC, courtesy of NASA Worldview.

## 3 Data & Methods

### 3.1 Observational Data

In this study, we use the International Best Track Archive for Climate Stewardship (IBTrACS) dataset as observational dataset (Knapp et al., 2010). This best-track dataset includes 3-hourly longitude/latitude positions of the eye (°), 6-hourly 10-meter 1-minute average maximum sustained wind speeds (m/s), 3-hourly minimum mean sea level pressure (hPa), and 3-hourly 17.5 and 25.7 m/s wind radius (km) data, as well as flags for status (low, tropical storm, hurricane, extratropical, etc), Saffir-Simpson Hurricane Wind Category (1-5), among others. All wind speed measurements were converted to 10 minute sustained mean wind speed, following Harper et al. (2010):

$$V_{10min} = 0.93V_{1min} \tag{1}$$

Due to Ophelia's atypical position on the eastern side of the Atlantic, there were very few direct measurements and these are mostly derived from remote sensing data. Figure 1(a) shows the IBTrACS best-track for Ophelia.

### 3.2 Analysis data

Prior to running our regional model simulations, we first determine which (re)analysis dataset performs best in capturing Hurricane Ophelia, and can therefore be used as input for the regional model. For this, we compare three widely-used (re)analysis datasets, namely ERA5 (Hersbach et al., 2020), the ECMWF operational output from IFS, and GFS (National Centers for Environmental Prediction, National Weather Service, NOAA, U.S. Department of Commerce, 2015). The ERA5 dataset is based on hindcast data from IFS; the GFS dataset is based on output from the NCEP Operational Global Forecast System model. ERA5 and GFS both have a model resolution of 0.25°, and IFS has a model resolution of 0.1°. Comparing the three datasets shows that all three datasets are capable of tracking Hurricane Ophelia through the Atlantic Ocean (Fig. 2(a)). However, ERA5 and IFS substantially overestimate Ophelia's central pressure (or, in other words, substantially underestimate Ophelia's intensity) in the tropical phase, by maximums of 36 and 32 hPa respectively (Fig. 2(b)). At the same time, GFS more closely follows Ophelia's intensification and subsequent weakening, with maximum deviations in the tropical phase of 11 hPa. These findings eliminate horizontal grid spacing as the sole cause of a potential intensity underestimation; GFS and ERA5 share the same model resolution, but GFS is in this case performing better than ERA5. Instead, these results hint towards an artefact in the IFS model. However, exploring this option lies beyond the scope of this research. Based on these results, we will use GFS data as input data for our regional model RACMO (see Section 3.3).

### 3.3 Model

#### 3.3.1 RACMO Description

We use the Regional Atmosphere Climate Model 2.3 (RACMO) (van Meijgaard et al., 2012). This model was developed by the Royal Netherlands Meteorological Institute (KNMI) and has been used with success in studies in Western Europe and

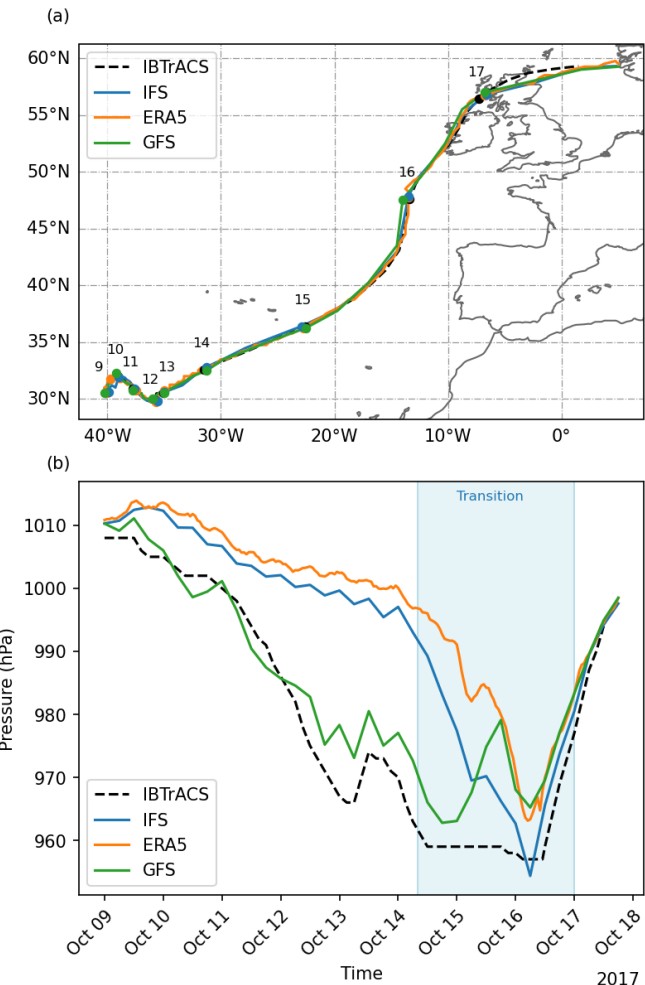

**Figure 2.** Track (a) and minimum central pressure (b) of Hurricane Ophelia for the ERA5, ECMWF Operational (IFS), and GFS datasets. Dashed line is IBTrACS observations. (a) Circles plotted at 00 UTC of the date indicated. (b) shaded area encompasses extratropical transition based on GFS simulation.

Greenland (Luu et al., 2023; Noël et al., 2015), as well as for TCs in the Caribbean (Dullaart et al., 2024). RACMO is an uncoupled hydrostatic atmosphere model with a horizontal grid spacing of 12 x 12 km, 40 hybrid sigma levels, and model physics based on ECMWF IFS Cy33r1 but an adjusted boundary layer scheme (Saarinen, 2004; van Meijgaard et al., 2012). Greenhouse gas concentrations are taken from CMIP5 simulations for October of 2017, and remain the same for all simulations (Taylor et al., 2012). More detailed specifications can be found in Appendix B and Table B1.

The initialization time for the RACMO simulations is 12 October 2017 00 UTC. We initially ran six RACMO simulations of Ophelia with initialization times between 09 – 14 October 2017, each at 00 UTC. The tracks and central pressure profiles of these simulations are shown in Figure B1.

The simulations initialized on 9, 10, and 11 October were not able to capture the observed strengthening of Ophelia as a hurricane and had quite large track deviations from the IBTrACS best track. The simulation initialized on 13 October strength-
125 ened Ophelia rapidly, surpassing the observed pressure values substantially, but also showing large track deviations. The 14 October simulation had a more reasonable track and smaller pressure deficit, but due to its late start it would not be possible to examine the ETT properly. As such, we chose the 12 October initialization. This is quite close to the 12 October 2017 13 UTC initialization time chosen by Rantanen et al. (2020) when they modeled Ophelia, citing similar challenges regarding the lack of strengthening.

In this study we examine the response of Hurricane Ophelia to alternate climate conditions. Ideally this would involve simulating only Ophelia, however to do so would require pulling the storm entirely out of its context. This is because it interacts with the jet stream and they together become one integrated system. This complicates the scenario, and so to keep our experiments as simple as possible we do not apply the full effects of a pseudo-global warming (PGW) approach. In an attempt to minimize displacement of the systems, we apply a spatially uniform $\Delta T$ to current climate forcing of both the atmosphere
and the sea surface temperature. This prevents temperature gradients that could shift the jet north or south of its observed position. While this is an approximation, it is one that is included in order to minimize the spread of possible outcomes such as the jet stream never interacting with Ophelia and its ETT never occurring at all. Our aim is to be able to draw conclusions on how climate change would specifically affect Ophelia. Additionally, we keep the relative humidity (RH) constant, which is consistent with global climate model study findings (Colman and McAvaney, 1997; Held and Soden, 2000, 2006).

We simulate seven alternate climate scenarios: starting with a control simulation (applied temperature forcing of 0°C) we simulate two scenarios of a cooler climate ( -2°C and -1°C) and four scenarios of a warmer climate (+1°C, +2°C, +3°C, and +4°C).

The same seven scenarios are simulated for four other initialization times (11, 13, 14, and 15 October, all 00 UTC) to examine the effects of initialization time on our results and if our signals are robust. These can be found in Appendix A.

### 3.4 Cyclone Tracking and Footprint

Unlike in studies such as Baatsen et al. (2015); Baker et al. (2024); Bourdin et al. (2024); Lopez et al. (2024) where large datasets with many TCs across decades are used, our single case allows us to use a simple cyclone tracking algorithm to find the location of Ophelia at each timestep. First the IBTrACS latitude/longitude position of the eye at the start of the simulation

is used as a guess for the initial position of Ophelia. The algorithm then searches the MSLP field in an 8° x 8° box centered on that guess for the local MSLP minimum. This location then becomes the position guess for the next timestep, and the process is repeated throughout the rest of the dataset. The IBTrACS data is used as an initial guess to prevent the algorithm from tracking other low pressure systems also present in the simulation, but is not used beyond that so as to avoid influencing the chosen track.

To determine the simulation-length spatial extent of strong winds, we examine the storm's wind footprint. Due to the presence of another low-pressure system in the North Atlantic before Ophelia, there is a substantial area of high winds not associated with the hurricane. To remove the influence of this and potentially other outside sources of high winds, we extract the 10m winds in a 30° x 30° box centered on Ophelia's eye latitude/longitude at every time step. The maximum wind over the simulation is found for each grid cell, and the number of gridcells exceeding the threshold value is multiplied by the area of one gridcell ($144 \text{ km}^2$) to find the total extent of the strong winds.

## 3.5 Quantifying Extratropical Transition

Observations showed that Ophelia had completed ETT and was a PTC starting from 16 October 00 UTC (Stewart, 2018). However, this does not tell us when ETT started nor how it progressed. Knowing where the storm is in its transition is an important aspect of predicting what the impacts will be. We employ several techniques which explore the conditions and timing of the ETT start and end, as well as what occurs during the transition, as the exact progression of ETT varies with every storm. In this paper we use cyclone phase space analysis and isobaric height gradient profiles.

Cyclone phase space (CPS) analysis is a technique used to examine the structure of TCs and ETCs, as well as quantify ETT. Hart (2003) introduced these now eponymous diagrams, and they have been used with success in many studies (Arnott et al., 2004; Dekker et al., 2018; Haarsma et al., 2013; Jones et al., 2024; Kitabatake, 2011; Kofron et al., 2010; Wood and Ritchie, 2014; Zarzycki et al., 2017). As this method solely uses the geopotential height on pressure levels between 300 and 900 hPa, it is relatively easy to adapt to different situations and datasets. Here we offer a brief explanation and we direct readers to Hart (2003) for more details.

CPS diagrams examine two main features that change during ETT: the horizontal thermal asymmetry ($B$) and the thermal wind ($-V_T$), which is divided into upper thermal wind ($-V_T^U$) and lower thermal wind ($-V_T^L$).

$B$ (m) is officially defined in Hart (2003) as "the storm-motion-relative 900–600 hPa thickness asymmetry across the cyclone within 500km radius":

$$B = h(\overline{Z_{600\,hPa} - Z_{900\,hPa}}|_R - \overline{Z_{600\,hPa} - Z_{900\,hPa}}|_L) \tag{2}$$

where $h$ is the sign to account for hemispheric differences (+1 for the Northern Hemisphere, -1 for the Southern Hemisphere), $Z$ is the isobaric height (m), $_R$ and $_L$ indicate the halves to the right and left of the storm respectively (relative to the direction of storm propagation). Horizontal thermal asymmetry is nearly non-existent in a TC ( $B \approx 0$ ) due to their radial profile with a warm center and cool exterior. In an ETC this is not the case, as the frontal nature of the storm means one side will be

significantly cooler than the other and thus geopotentially thinner. Evans and Hart (2003) determined that for North Atlantic storms, ETT starts when the asymmetry parameter $B$ is greater than 10 m.

The upper and lower thermal wind are calculated as:

$$-V_T = \frac{\partial(\Delta Z)}{\partial(\ln p)} \tag{3}$$

where $p$ is pressure and $\Delta Z$ is the height perturbation on that pressure level ($\Delta Z = Z_{MAX} - Z_{MIN}$). The thermal wind can be used as an approximation to the cyclone isobaric height gradient, which is the height perturbation on a pressure level, $\Delta Z$, in a 500 km radius from the center of the cyclone. This is related to the lower-troposphere thermal wind, as a warm core system will have a positive $-V_T^L$ because the perturbation at 900 hPa would be larger than at 600 hPa.

Positive values of the thermal wind ($-V_T$) indicate a storm core that is warm compared to the environment, and negative values indicate a storm core that is cold compared to the environment. This relationship allows us to study ETT, as generally TCs have warm cores and ETCs have cold cores.

Relatedly, the second analysis method we use is an isobaric height gradient profile. We quantify the end of ETT by following the definition of Evans and Hart (2003): ETT is completed when the 900-600 hPa cyclone isobaric height gradient corresponds to that of a cold-core storm (i.e., increasing with height) rather than a warm core (decreasing with height). This cold-core isobaric height gradient is also visible as a negative value of the thermal wind ($-V_T^L$).

$$-V_T^L = \frac{\partial(\Delta Z)}{\partial(\ln p)}\bigg|_{900hPa}^{600hPa} \tag{4}$$

In Hart (2003) a 50 hPa vertical interpolation is used, but due to the lack of pressure levels in our data, we calculate $-V_T$ as a simple difference between the upper and lower bounds (300 and 600 hPa for $-V_T^U$ and 600 and 900 hPa for $-V_T^L$). As we do not have data at 600 or 900 hPa, these are calculated as linear interpolations between 500 and 700 hPa and 850 and 925 hPa respectively.

The calculated $B$, $-V_T^U$, and $-V_T^L$ data are smoothed using a 12-hour convolution filter to remove noise present due to the chaotic nature of the track, where slight shifts can change the angle of propagation and thus the area over which these variables are calculated. Hart (2003) used a 24-hour running mean filter for the same purpose, however due to the higher temporal resolution of our simulations (hourly vs their 6-hourly) we use a shorter interval.

## 4 Results

### 4.1 Downscaling with RACMO

First, we examine how RACMO models Ophelia in the downscaled GFS analysis data. Looking at the $0°C$ simulation initialized 12 October 2017 00 UTC, we see that Ophelia's track is quite well simulated (see Fig. 3). There is a slight deviation before the storm hits land: where IBTrACS and the GFS have Ophelia continue turning north for longer before heading more sharply east, in the RACMO simulation the storm keeps a more consistent heading. The landfall location is nearly the same, though the angle becomes more oblique to the coast.

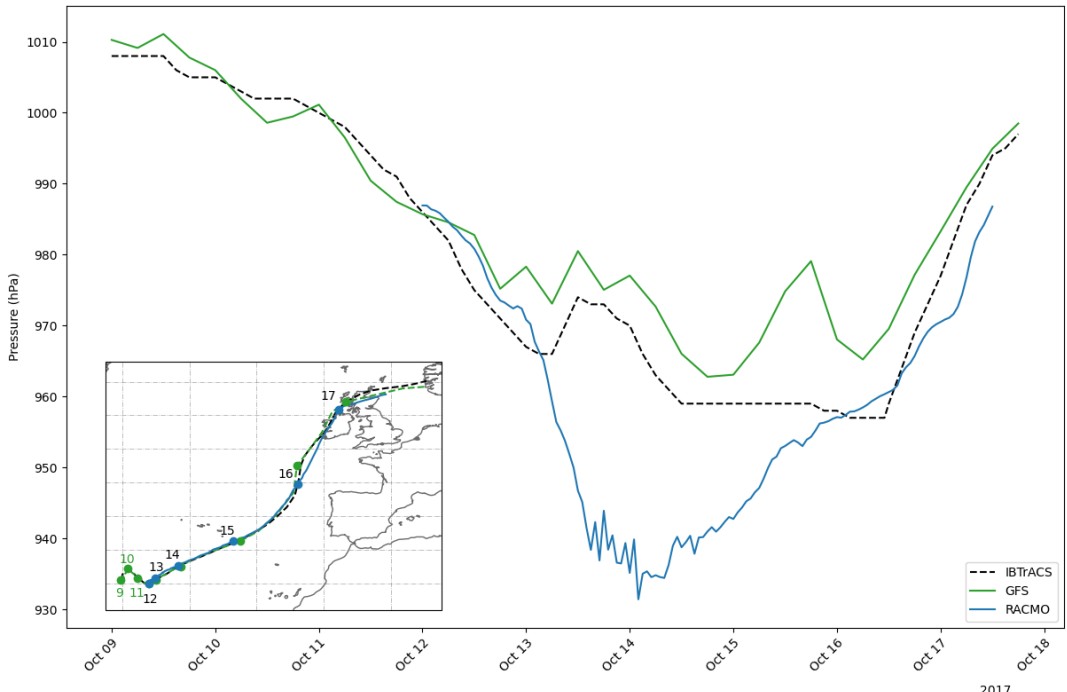

**Figure 3.** Minimum central pressure profile of Hurricane Ophelia for the RACMO 0°C simulation initialized 12 October 2017 00 UTC (blue) compared to the GFS simulation (green) and the IBTrACS observations (black). Inset figure shows the associated tracks with daily storm locations.

The intensity of the storm is less well simulated, with a much lower central pressure in the RACMO simulations, overshooting the minimum IBTrACS and GFS pressures by 26 and 32 hPa respectively. Despite this, we chose to use this initialization time over another that did not capture Ophelia's strengthening as a TC (see Sect. 3.3).

## 4.2 Alternate climate scenarios

Ophelia's tracks in the alternate climate simulations (Fig. 4) initially show high levels of lateral clustering, but after 15 October they diverge. There is a clear and systematic response to warming: simulations with higher temperature forcing place Ophelia further northwest than those with lower temperature forcing. This difference occurs as a result of an earlier, stronger northward turn and peaks at 15 October 23 UTC, with Ophelia being 559 km further northwest in the +4°C simulation than in the -2°C simulation. This is an average of about 90 km further northwest per degree of warming. The three warmest simulations (+2°C, +3°C, and +4°C) even have Ophelia not making landfall in Ireland, but passing by off the west coast. Accompanying these track changes is also a difference in translation speed, with higher temperature forcing scenarios having higher translation speeds (see Fig. B4). This is in line with findings by Sun et al. (2021) who have found that TC translation speed has increased in the North Atlantic in the period 1982-2016. Note that since our alternate climate approach did not include large-scale circulation

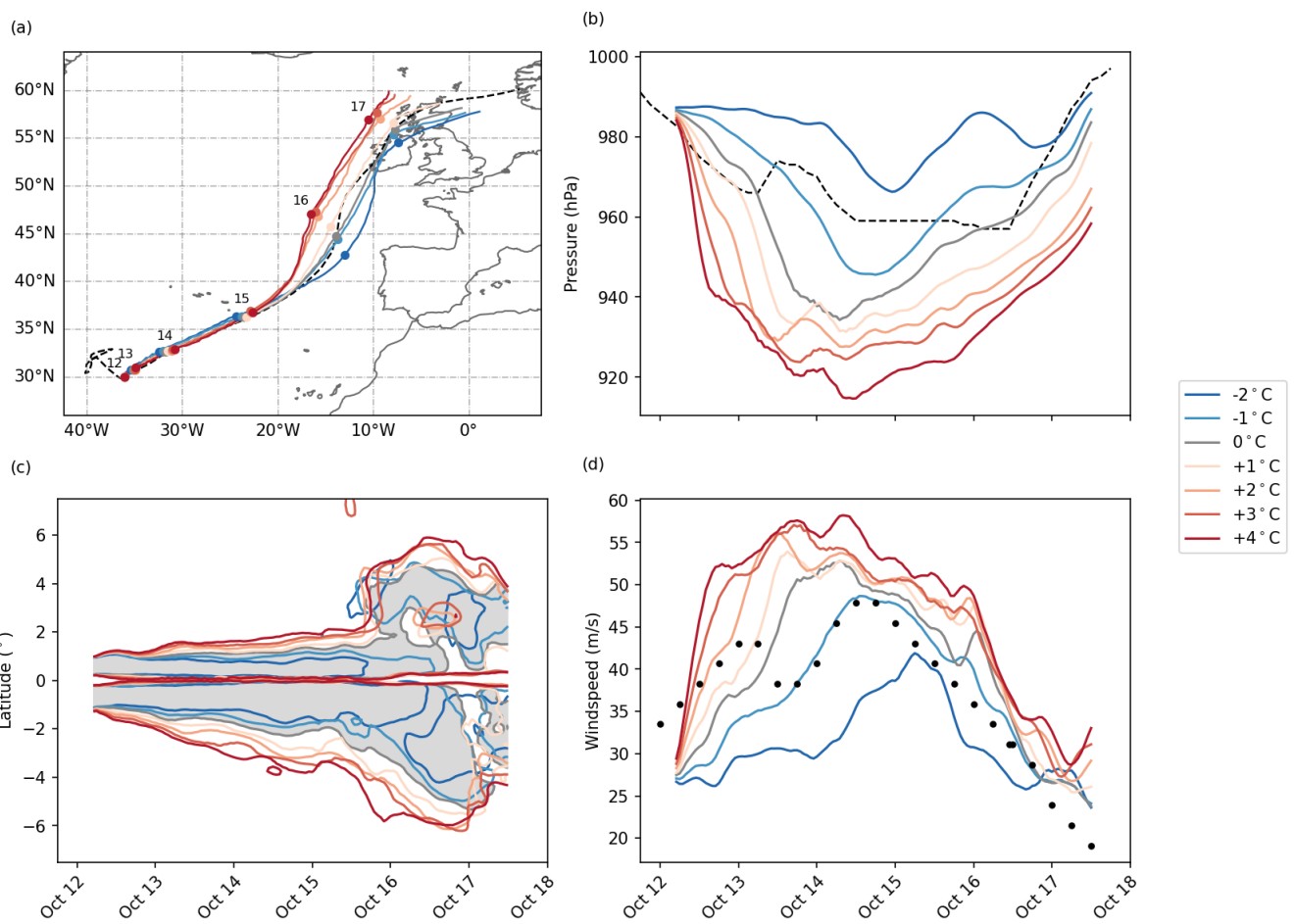

**Figure 4.** Track (a), minimum central pressure (b), north-south diametric slice of extent of 10-minute sustained 17 m/s wind, in degrees relative to storm centre (c), and 10m instantaneous maximum wind (d) of Hurricane Ophelia for the alternate climate downscaled RACMO simulations with GFS boundaries, initialized at 12 October 2017 00 UTC. Dashed line (black dots in (d)) are IBTrACS observations. (a) Circles plotted at 00 UTC of the date indicated. Shading in (c) is the extent of the 0°C simulation. A 6-hour convolution was applied to panels (b) through (d) to filter out noise.

**Table 1.** Minimum central pressure, maximum central pressure deviation compared to IBTrACS observations, and maximum 10m wind speed for each alternate climate scenario.

| Temperature Forcing (°C) | Minimum Central Pressure (hPa) | Maximum Central Pressure Deviation (hPa) | Maximum 10m Wind Speed (m/s) | First 12 h Wind Speed Change (m/s) |
|:---:|:---:|:---:|:---:|:---:|
| -2 | 965.7 | +28.1 | 43.0 | +0.7 |
| -1 | 944.9 | -15.0 | 49.2 | +6.4 |
| 0 | 931.4 | -34.9 | 54.8 | +9.5 |
| +1 | 928.7 | -43.3 | 55.4 | +14.2 |
| +2 | 926.5 | -44.0 | 57.7 | +21.4 |
| +3 | 920.7 | -46.5 | 59.4 | +23.3 |
| +4 | 912.6 | -51.0 | 59.4 | +25.2 |

changes, these track changes *must* be due to (thermo)dynamic adjustments in the internal dynamics of Ophelia, the adjustment of the jet to the uniform warming or cooling, or the interaction of Ophelia with the jet. Below we consider these aspects in more detail.

### 4.2.1  Internal dynamics adjustment

The main adjustments that occur in the internal dynamics of Ophelia as a result of the imposed heating is that the storm
increases in both intensity and size. The higher SSTs provide more fuel for the convection that powers the storm in the form of latent heat release. This allows for an intensity increase that can be identified by examining the central pressure or maximum wind speed. Ophelia exhibits a strong, nearly-linear trend in central pressure minima: in scenarios with higher temperature forcing Ophelia has a lower central pressure (Fig. 4(b)), implying a stronger storm. The +4°C simulation has the lowest central pressure minimum at 913 hPa, while the -2°C has Ophelia's highest central pressure minimum at 966 hPa (Table 1).
Additionally we see several interesting phenomena in these results. Firstly, there is a misalignment of expected pressure and wind speed changes. Despite there being a large decrease in central pressure compared to the observations, we do not see a comparable increase in wind speed. Ophelia's estimated maximum sustained 10 minute wind speed was 47.8 m/s at a central pressure of 959 hPa (Stewart, 2018). Our most extreme simulation of Ophelia (+ 4°C) with a minimum pressure of 912 hPa has a maximum 10m wind speed of 59.4 m/s, making it a mid-range Category 4 storm. This wind speed was reached on 14
October 05 UTC, when Ophelia was still a TC (see Section 4.3), though these central pressures are more often associated with Category 5 storms, which indicates there are probably limitations due to the model resolution. The 12 km model resolution is quite low compared to eye sizes/radius of maximum wind ($R_{max}$) values, which can cause underestimation of storm intensity (see Fig. B3(a), Bloemendaal et al. (2019)). Such an underestimation of TC intensity compared to central pressure has been noted in multiple studies, where the 10m wind is generally underestimated by models, indicating that this may be a more far-
reaching problem than just in this study(Michaelis and Lackmann, 2019; Murakami et al., 2015; Roberts et al., 2015; Yamada

et al., 2017) In addition, Fig. 4 also reveals that the maximum wind speeds occurring after 16 October 06 UTC and before reintensification at or after 17 October 06 UTC are all very similar (with a maximum difference of 11.4 m/s between the -2°C and +4°C scenario), despite large differences in central pressure (of up to 47.7 hPa between the -2°C and +4°C scenario). The similarity in wind speeds here may be as a result of the storm interacting with land, and a decrease in pressure gradient. The
storms in warmer scenarios are larger than those in cooler simulations, so even with deeper central pressure they could have similar pressure gradients and thus similar wind speeds (see Figure B5). Additionally, interaction with land could be a factor in the decreasing wind speeds as despite their more westerly track, the warmer storms come into contact with the land earlier than the cooler storms (not shown).

Secondly, the changes in intensity of the storm vary greatly between simulations. The initial intensity of the storm increases
much more rapidly in warmer simulations than in cooler simulations. The maximum wind speed of the +4°C simulation increases by 25.2 m/s in the first 24 hours (taking it from a tropical storm to the edge of a Category 3/4 storm in that time), with successively smaller increases ending with the -2°C simulation adding only 0.7 m/s in the same time period (see Table 1). We see that all but the -2°C and -1°C simulations eventually undergo rapid intensification (RI), which is defined as a >14.0 m/s increase in 10 minute sustained wind speeds in 24h (Kaplan and DeMaria, 2003), though this takes longer in the less strongly
forced simulations. In the observations of Ophelia there is no evidence of rapid intensification, which may be due to a period of shear-induced weakening that RACMO does not model.

Thirdly, the central pressure of the storms exhibits rapid oscillations in the first half of the simulations. This is likely attributable to the storm reaching the edge of the model capability, as it is accompanied with a decrease in eye size (as visible in 4(c)) and a decrease of $R_{max}$ (see Fig. B3(a)). With the complicated system of updrafts, subsidence, and other sharp gradients
present within the eye, it is not possible to fully model this with the current resolution. Thus the model relies on parametrization for these sub-gridscale processes, which due to the intense nature of the storm may mean the model approaches the limits of these parametrizations, and thus has some difficulty capturing the true structure and thus strength of the storm.

Lastly, the coldest simulations (-2°C and -1°C) provide insight into the nature of the storm. In the scenarios simulating a colder-than-current climate, Ophelia's tropical phase is much weaker or even not present. This suggests that the tropical
evolution of Ophelia was at the edge of what was possible climatologically and was strongly dependent on the above average warmth of the SSTs under the storm in combination with cool upper level temperatures (Rantanen et al., 2020). Additionally, the post-transition phase in these scenarios was substantially weaker than the warmer-than-current simulations, which demonstrates that the tropical origin of Ophelia plays an important role in the intensity and impacts of the extratropical storm.

Besides the previously-mentioned intensity increases, the warmer runs also show a storm size increase (see Fig. 4(c)), a
characteristic related to increases in the surface latent heat flux (Radu et al., 2014). This increase in storm size and intensity also increases the beta drift effect on Ophelia. The beta drift effect is the result of the interaction between the rotating storm and the planetary vorticity gradient, resulting in a tendency to advect the storm poleward and westward (DeMaria, 1985; Carr and Elsberry, 1997). Larger and stronger storms are more strongly affected by the beta drift effect than smaller, weaker storms. It follows that by its size the cyclone also actively modifies its own propagation, forcing the storm further to the northwest.

**Table 2.** Peak beta drift for Ophelia in each of the simulations, ranked.

| Ranking | Simulation | Peak beta drift (unconvolved) (m/s) |
|---------|------------|-------------------------------------|
| 1 | +4°C | 4.72 |
| 2 | -1°C | 4.44 |
| 3 | +3°C | 4.38 |
| 4 | +2°C | 4.18 |
| 5 | +1°C | 4.17 |
| 6 | 0°C | 4.14 |
| 7 | -2°C | 3.21 |

We compute the beta drift using Smith's Hurricane Beta-drift law (Smith, 1993):

$$BD = 0.72B^{-0.54}r_{max}^2\beta$$

where

$$B = \frac{r_{max}^2\beta}{V_{max}}$$

We find that the beta drift is generally higher for the storms in warmer simulations and lower for those in colder simulations, especially in the first 72 hours of the simulations (see Fig. B3(b)). The +4°C simulation has a 72-hr mean beta drift of 1.25 m/s, and the -2°C has a mean beta drift of 1.02 m/s. These are small changes compared to the 15 m/s translation speed the storms achieve around 15 October 00 UTC, but not negligible, especially closer to the beginning of the simulation where the storms are traveling slower; by 13 October 00 UTC there is already a visible difference in position between simulations (Fig. 4) but

the difference in translation speed between the fastest and slowest simulations (+4°C and -2°C respectively) is only 1.59 m/s.

When we examine this for the entire length of the simulation, we find that there is a stratification of the beta drift with temperature: in general, storms in warmer simulations having a higher $V_{max}$ and a lower $R_{max}$ show higher values of maximum beta drift. The +4°C has a maximum beta drift of 4.72 m/s, where the -2°C has a maximum beta drift of 3.21 m/s. With the exception of the -1°C simulation, the relationship of storms in warmer simulations experiencing stronger beta drift is maintained

(see Table 2). The high beta drift of the -1°C scenario can be attributed to an anomalously high $R_{max}$. In other words, the storms in warmer simulations propagate faster and thereby further northwest than those in cooler simulations.

### 4.2.2 Adjustment of the jet

The adjustment of the jet to the uniform warming is visible in the 200 hPa wind velocity (Fig. B2). Using prescribed boundaries and predominantly westerly inflow, changes in the wind velocity in the outer and western parts of the domain are negligible

throughout the length of the simulations. However, in the central part of the domain the jet stream intensity increases considerably, even before the interaction with Ophelia. This is likely an internal adjustment response of the model to the warming, as deeper convection warms higher layers of the atmosphere, increasing vertical shear and the jet stream speed. We also find

that the position of the jet changes little prior to the interaction with Ophelia, as is to be expected both based on the prescribed boundaries and similar conclusions drawn by other studies (Bond et al., 2010; Ren et al., 2014).

Behind Ophelia the jet stream stays consistent across simulations as it flows in from the west, with only slight deviations in extent of strong winds and maximum wind speed (Fig. B2). For much of the simulation the location of maximum 200 hPa wind is near the western boundary, which explains its consistence across simulations, as there has been little time to adjust to the altered conditions.

### 4.2.3    Interaction of Ophelia with the jet

The phasing of the storm-jet stream trough has impacts on the post-transition intensity change. Ophelia approaches the jet stream while located under the right entrance region of the jet streak (Fig. B2), which has been shown to favor cyclone intensification (Ritchie and Elsberry, 2007; Sarro and Evans, 2022). This occurs as a result of upper level divergence, which helps vent the upper levels of the storm, preventing the centre from "filling in" with boundary layer air.

    Once Ophelia and the jet stream start interacting, it is difficult to separate one from the other, and other similar studies have
started classifying them together as a system (Sarro and Evans, 2022). The interaction with the jet stream does not just affect Ophelia, but also the jet stream itself. The jet stream in the warmer scenarios has a higher speed (86.3 m/s in the +4°C vs. 66.8 m/s in the -2°C), but this is only really noticeable in the period between 13 October 12 UTC and 15 October 03 UTC (Fig. B2).

    As shown in Sect. 4.2.1 above, in scenarios with higher temperature forcing Ophelia becomes a larger, stronger storm. As a larger storm, Ophelia would be influenced by the jet stream sooner than had it been a smaller storm, which could explain
both the earlier movement we see in the higher temperature tracks (Fig. 4) and the higher translation speed in the warmer simulations (Fig. B4). Additionally, stronger TCs are in general more controlled by upper level flow than weaker TCs (Sun et al., 2021). Combining this with the stronger jet stream in the warmer scenarios noted in Sect. 4.2.2, the storm is provided with extra propulsion and the separation between warm and cold scenarios increases.

### 4.3    Extratropical Transition

Next, we examine the ETT of Hurricane Ophelia. Because the characteristics and thus the impacts of a TC differ from those of an ETC, the storm's ETT profile and timing directly affects the associated impacts. As such we examine the start, progression, and end of Ophelia's ETT through the use of the cyclone phase space method (Evans and Hart, 2003).

### (1) : Start

Based on the temporal evolution of the thermal asymmetry parameter $B$ (see Fig. 5 and the definition in Evans and Hart (2003)),
we can pinpoint when Ophelia's ETT starts in each of the alternate climate scenarios. In each scenario, $B$ initially remains well below the ETT threshold value of 10 m, as expected for a symmetric TC. All of the simulations cross the ETT threshold (starting their ETT) between 8 and 16 UTC on 14 October (5(b)). In general, the scenarios with lower temperature forcing start

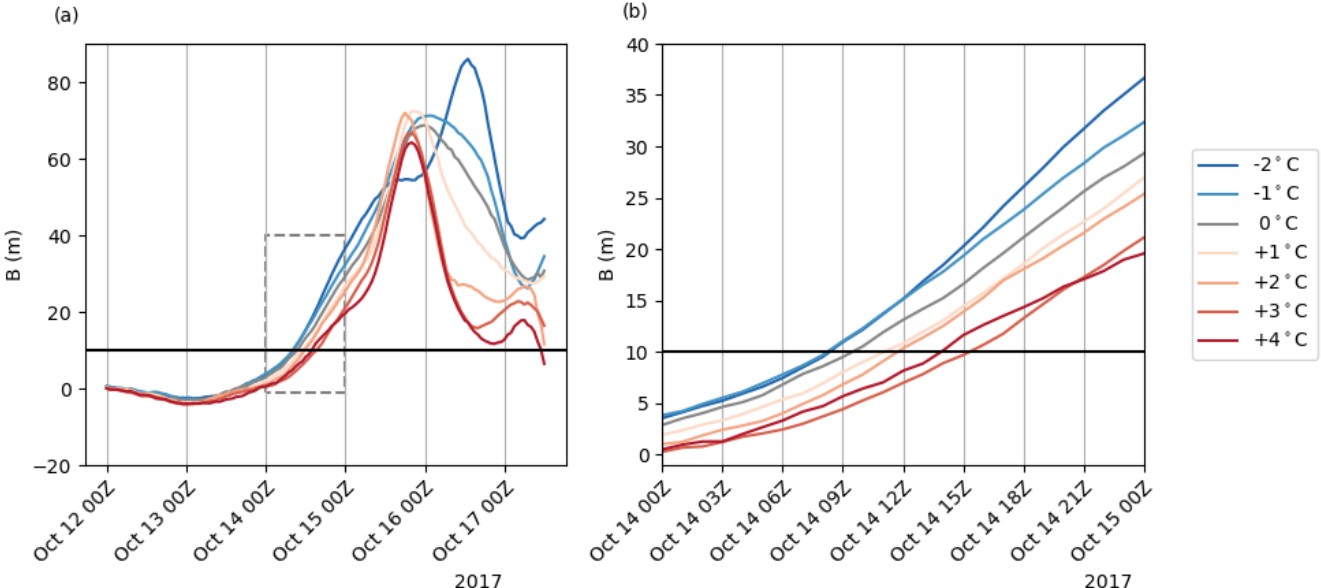

**Figure 5.** 900-600 hPa thickness asymmetry for the varying temperature scenarios for (a) the full length of the simulations and (b) zoomed in on the extratropical transition start. The outlined square in (a) shows the bounds of (b).

ETT earlier than the scenarios with higher temperature forcing, but there is some fluctuation, with the +4°C simulation leading the +3°C, and the -2°C and -1°C simulations starting their ETTs at nearly the same time.

**(2): Progression**

The asymmetry can also tell us about the progression of the ETT: the warmest simulations become less asymmetric overall than their cooler counterparts (see Fig. 5 (a)), with the +4°C achieving only 75% of the asymmetry of the -2°C simulation (64.2 m vs 86.0 m).

The fact that in the warmer simulations Ophelia remains thermally symmetric for longer means that it retains its TC charac-
teristics longer than in cooler simulations. This can be attributed to the storm having a greater ability to favourably condition its environment in the warmer simulations. In the plots of 15 October in Figure 6, the storms in the warmer simulations have a larger envelope of positive geopotential height anomaly (shown in red in Figure 6), indicating warmer than average air, than the cooler simulations. A larger envelope forms a buffer of sorts to the cooler and geopotentially thinner air (shown in blue in Figure 6) that would increase the asymmetry of the storm. Additionally, the warm core itself (seen in the intense reds in 6) is
both larger and stronger in the warmer cases. The seemingly decreasing average anomaly on 16 October 12 UTC is due to the varying values of the latitudinal average used. In cooler simulations, this average tends to be lower.

All simulations except the -2°C have a peak in asymmetry at the end of 15 October, after which their asymmetry decreases again. The three warmest simulations very quickly attain low levels of asymmetry and return to near-TC levels. These results

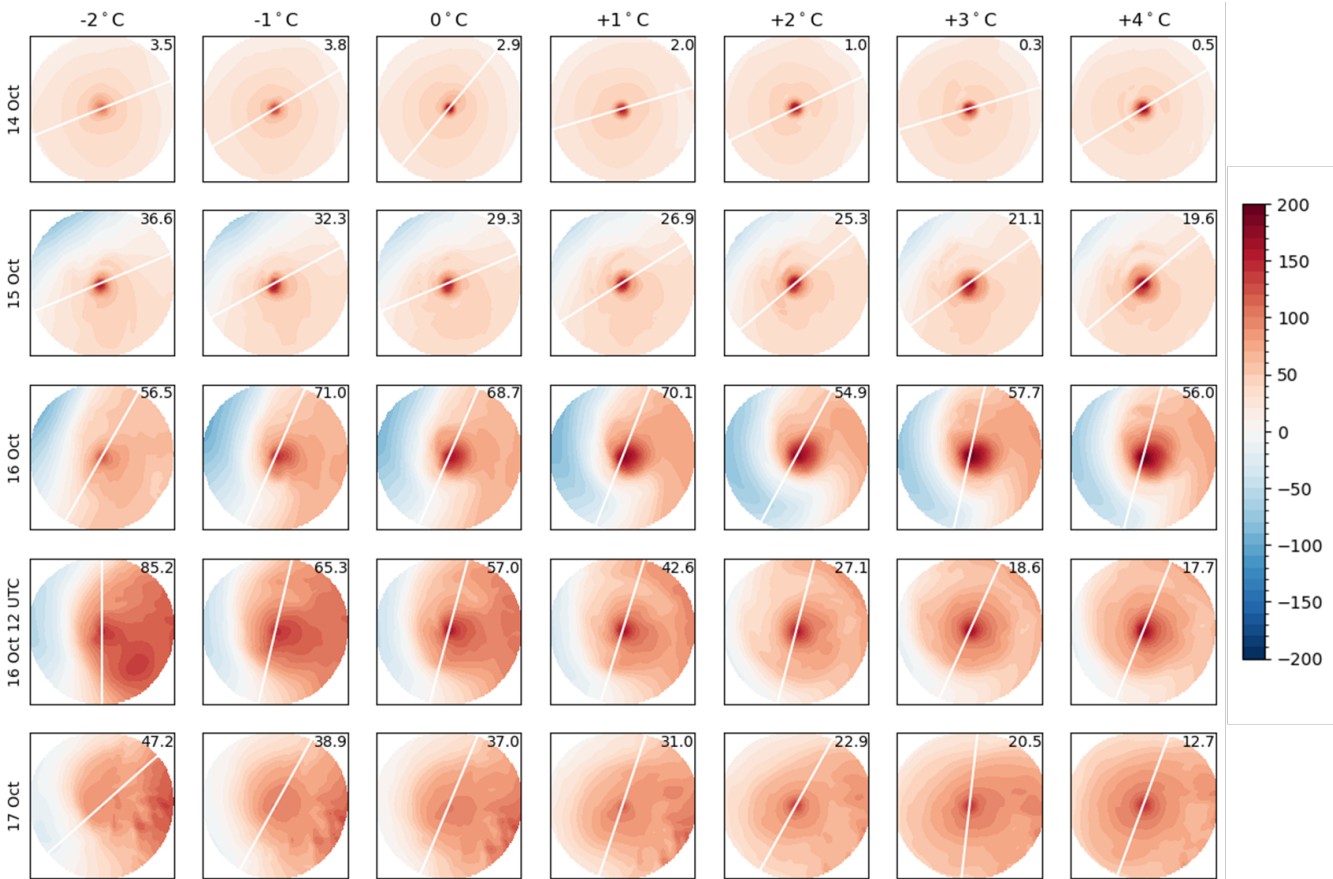

**Figure 6.** 900-600 hPa geopotential height difference anomaly (m) compared to scenario-specific simulation-length average of the area +/- 5° around the storm's latitude removed, for circles of 500 km radius around the storm for several timestamps and each temperature scenario. The number in the top right is the value of the thermal asymmetry parameter for that timestamp and scenario. The white line shows the storm translation direction, as determined by a centered difference of the storm track.

are indicative of an "instant warm seclusion", a term coined by Sarro and Evans (2022), to describe a storm that immediately

attains a storm structure similar to that of a warm seclusion cyclone but without going through the cold-core structure typical of that pathway. As the warmer scenarios return to TC-like faster and with less asymmetry than the cooler simulations, this implies that Ophelia maintains more of its TC-like characteristics in the warmer scenarios. This progression can also be seen in Figure 6: while all scenarios undergo the transition from radially symmetric TC on 14 October to asymmetrical on 16 October, we see that by 16 October 12 UTC the storms in the warmest scenarios return to the radially symmetric structure more than

the cooler simulations.

We can put this progression of ETT in more context by looking at the full CPS diagrams for Ophelia (see Fig. 7(a)). The simulations with higher temperature forcing have a higher $-V_T^L$ than the simulations with lower forcing, indicating they have

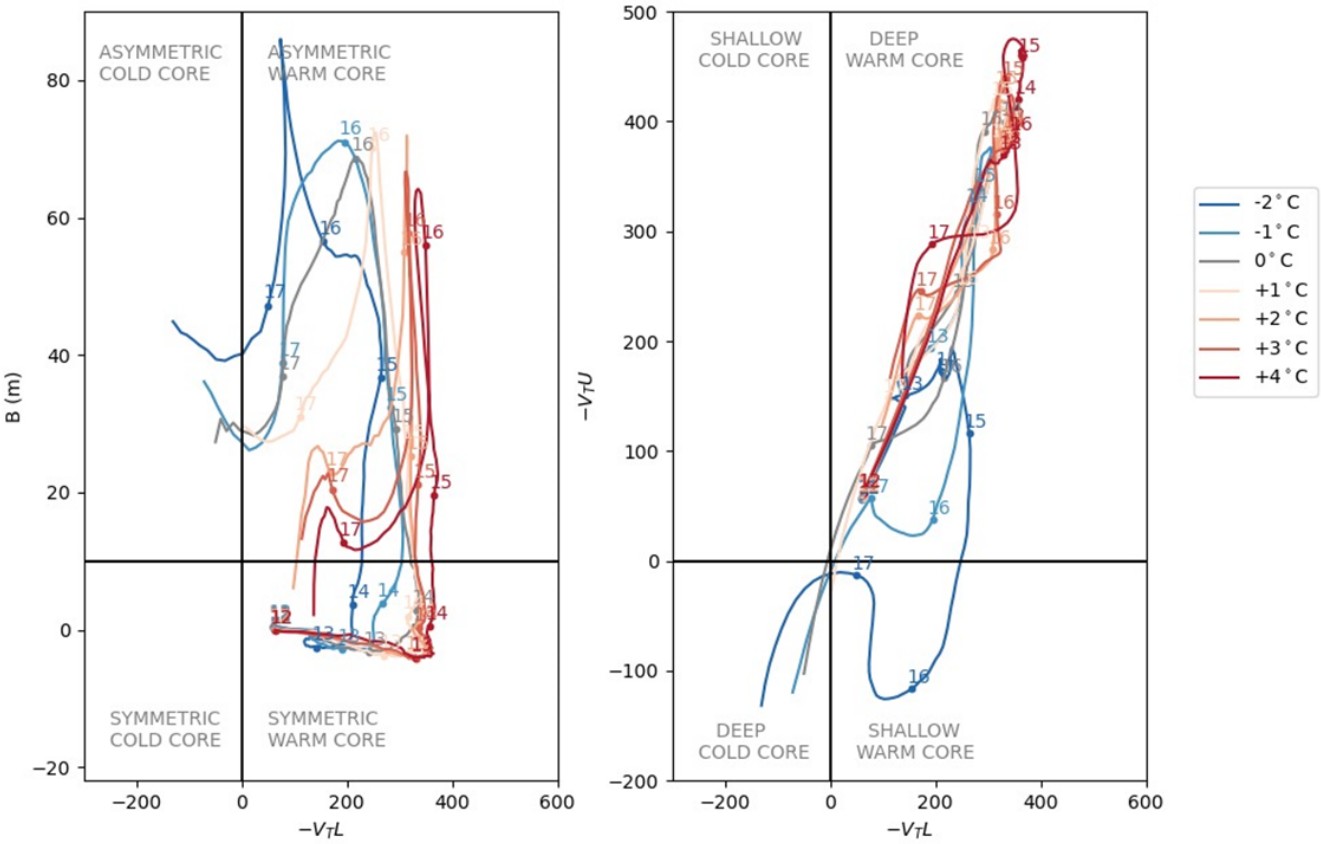

**Figure 7.** Hart Diagrams for the RACMO alternate climate scenarios, with (a) lower tropospheric thermal wind $(-V_T^L)$ vs thickness asymmetry ( $B$ ) and (b) lower tropospheric thermal wind $(-V_T^L)$ vs upper tropospheric thermal wind $(-V_T^U)$.

stronger warm cores in the lower troposphere. This is to be expected with the $\Delta$T we introduced, as the higher SSTs mean more available latent heat, and thus more warming in the atmosphere due to latent heat release. The +2°C,+3°C, and +4°C

temperature forcing maintain their $-V_T^L$ after reaching peak asymmetry and $-V_T^L$ only starts decreasing substantially on 16 October, with warmer simulations starting this later than cooler simulations. In contrast, simulations with +1°C or less temperature forcing start decreasing their $-V_T^L$ before or at peak asymmetry.

  The simulations with higher temperature forcing have higher $-V_T^U$ as well (Fig. 7(b)). This suggests that Ophelia's convection in the warmer scenarios is stronger, because more heat is making its way up to a higher level of the atmosphere. We

also observe the same maintenance of $-V_T^L$ after 15 October noted in Fig. 7a, but now it is coupled to a slight decrease in $-V_T^U$ instead of the decrease in asymmetry. There are two processes at work here simultaneously: continued latent heating of the lower troposphere and decreased heating per unit area in the upper troposphere. The lower troposphere can continue maintaining its $-V_T^L$ through latent heat release even after ETT, which helps maintain the warm-seclusion (Quitián-Hernández et al., 2020). Meanwhile the tilting of the storm column with height due to the cooler (geopotentially thinner) air to its west

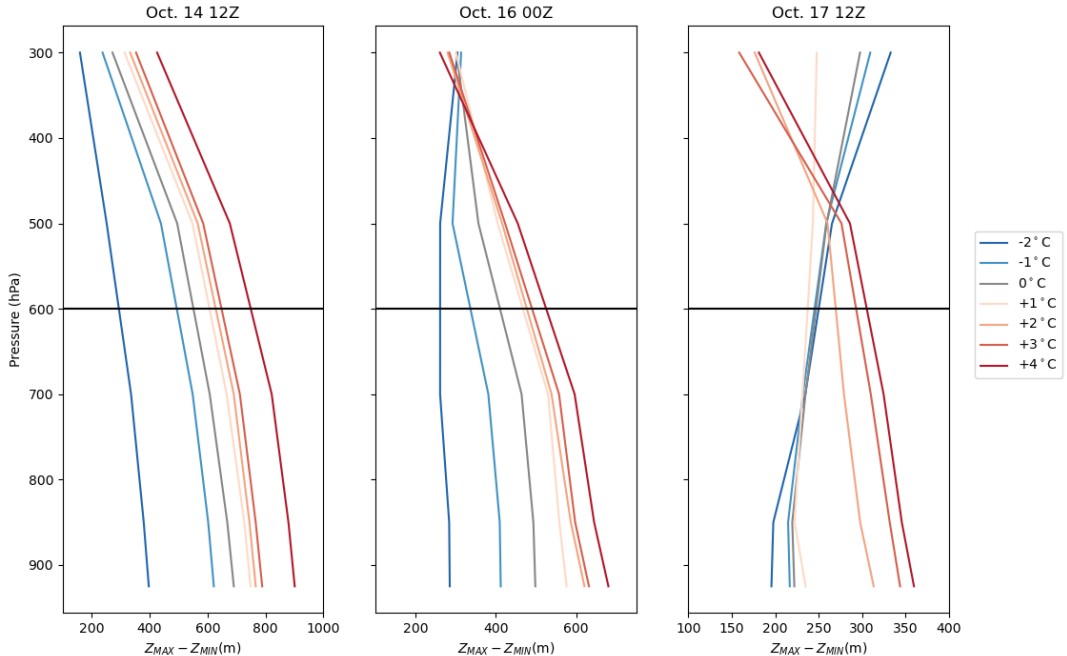

**Figure 8.** Isobaric height gradient profiles for each temperature scenario for the approximate start of extratropical transition (14 October 12 UTC, left), at the National Hurricane Center-determined extratropical transition completion point (16 October 00 UTC, middle), and at the end of the simulations (17 October 12 UTC, right). Please note the different x-axis ranges.

increases the volume of air in the core, decreasing the per unit volume heating even with the same input of energy. However this heating also starts to decrease as Ophelia moves over colder water. We can also attribute the eventual decrease of both $-V_T^U$ and $-V_T^L$ to the lower-SST-driven decrease in available latent heat.

**(3): End**

To quantify the end of Ophelia's ETT we use Eq. 4 and examine the evolution of Ophelia's isobaric height gradient profiles in
each alternate climate scenario (see Fig. 8 and Sect. 3.5 for explanation).

     At the approximate start of the ETT (Fig. 8, left panel) Ophelia is still a warm-core storm in all scenarios, as the profiles of the storm show decreasing $Z_{MAX} - Z_{MIN}$ with height, specifically for pressure levels under 600 hPa. We also see a clear ordering of $Z_{MAX} - Z_{MIN}$ magnitude, with warmer simulations showing strictly greater $Z_{MAX} - Z_{MIN}$ values than colder simulations.

Observations indicated that Ophelia had completed ETT by 16 October 00 UTC (Stewart, 2018). We see that our simulated versions of Ophelia lag behind this however. At 16 October 00 UTC in the -2°C scenario, Ophelia leans towards a cold-core profile in the upper levels, but it maintains a neutral profile in the lower troposphere which would indicate ETT is ongoing but not yet completed (Fig. 8, middle panel). The convection in the -1°C and 0°C scenarios is also starting to become shallow but

the rest of the simulations are still strongly warm core, which indicates ETT has not fully taken hold. The overall $Z_{MAX} -$
$Z_{MIN}$ magnitude has also decreased, and in the upper portion of the atmosphere we see the tilt reversal taking place as the storm adopts more ETC-like characteristics in the colder scenarios. The stronger jet stream noted in the warmer scenarios in Section 4.2.2 may also play a role in this: the greater wind shear associated with a faster jet core could more effectively ventilate the upper levels of the storm.

At the end of our simulation time (17 October 12 UTC, Fig. 8, right panel), the +2°C through +4°C simulations are still strongly warm-core, the -2°C through 0°C have become cold core, while the +1°C exists as a hybrid between warm-core (TC-like) and cold-core (ETC-like) storms. The persistent deep warm-core in the simulations with warming suggest that Ophelia does not finish its ETT but remains a hybrid form of a TC and ETC.

## 4.4 Impacts

In the previous section, we demonstrated that in the warming scenarios Ophelia does not finish ETT but instead remains a warm-core hybrid form of TC and ETC in the warmer scenarios. Such hybrid storms pose a potential threat to areas like Ireland which are more used to ETCs than TCs. TCs bring a different structure and impact footprint than ETCs: in general, TCs are stronger (in terms of wind speed) storms than ETCs. TCs also have only slight wind field asymmetry due to the influence of lower translation speed and vertical wind shear. These influences, however, are larger in ETCs which therefore show larger wind field asymmetries (Jones et al., 2003).

PTCs can display a mixture of TC and ETC characteristics, especially while they are still transitioning from one to the other. As they undergo ETT, the radius of maximum wind increases and the entire wind field expands and becomes more asymmetric (Evans et al., 2017). Their translation speed also often increases as a result of interaction with upper level jets, which adds to the wind field to produce stronger wind speeds even as the pressure-induced wind field weakens (Hart and Evans, 2001). As such, PTCs can bring high wind speeds similar to TCs over a large area like ETCs, increasing the potential damages.

The overall maximum 10m sustained wind speed increases with increasing temperature forcing (from 43 m/s to 59 m/s in scenarios -2°C to +4°C, see Fig. 9), which is consistent with the strengthening storm we see in the decreasing central pressure (Fig. 4). The area affected by Ophelia's winds increases simultaneously, with both the storm-force (10 Beaufort (Bft), $\geq$ 24.7 m/s) and hurricane-force (12 Bft, $\geq$ 32.8 m/s) winds increasing substantially: the +4°C simulation has, respectively, areas 157% and 162% larger compared to current climate conditions (0°C; see Fig. A3).

Interestingly, although the track moves westward with warmer scenarios, the winds experienced around Ireland are stronger than in the scenarios where the storm hits the country directly (Fig. 9). This is predominantly caused by the combination of Ophelia becoming more powerful and its wind field expanding in the warmer runs compared to the control run.

## 5 Discussion

Dullaart et al. (2024) used RACMO to downscale TCs in the Caribbean using RACMO under PGW conditions. They found that while RACMO is able to model the number of TCs relatively well, it does not achieve the same intensity distribution.

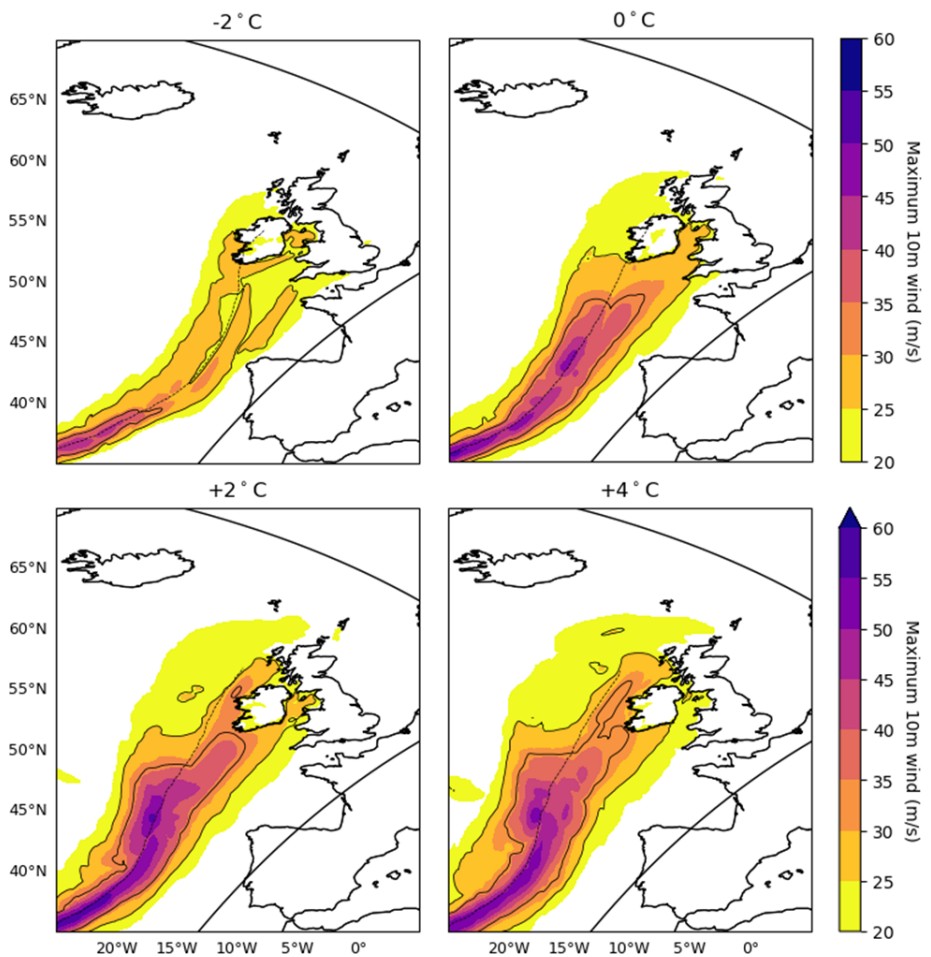

**Figure 9.** 10m wind footprint for Ophelia in the -2, 0, +2, and +4°C alternate climate scenarios, with their respective tracks. Contour lines at storm-force (10 Bft, 24.7 m/s) and hurricane-force (12 Bft, 32.8 m/s). RACMO domain bounds are also shown.

RACMO intensifies more storms to around Category 3/4 than we see in the observations, and underestimates both the number and relative frequency of maximum wind speed of both ends of the scale: Category 5 storms and Category 1 and below are underrepresented. This is reflected in what we see in our simulations of Ophelia: all of our storms reach maximum wind speeds of Category 3 or 4 despite massive differences in central pressure (see Table 1). Dullaart et al. (2024) also find a substantially larger percentage of storms undergo RI in their RACMO simulations than in the observations (71% vs 91%), and they attribute this to RACMO being an uncoupled model with no cold upwelling feedback. This could also help explain why we see simulations +0°C through +4°C undergo RI even though this does not occur in the observations.

Sarro and Evans (2022) investigated a series of North Atlantic TCs that underwent ETT, and among others identified Ophelia as an "instant warm seclusion" type of transitioning TC. These storms differ from more traditional, "post-transformation" warm

seclusion storms in that they do not first achieve cold-core status before returning to warm core status; rather, these storms never achieve cold core status at all. We see evidence of this also being the case for our simulations of Ophelia, with the lack of cold-cores in nearly all simulations (see Fig. 8). This is in line with Baatsen et al. (2015)'s findings for both their case study storm Amy and other near-future PTCs. While the magnitude of $Z_{MAX} - Z_{MIN}$ has clearly decreased with simulation time, none of the simulations truly finish their transition and become a cold-core storm, staying this hybrid of TC and ETC.

The longer maintenance of high $-V_T^L$ in the warmer simulations may be the warm core of Ophelia, in its transition to an ETC, still taking up latent heat from the high SSTs (heating diabatically). Rantanen et al. (2020) found that diabatic heating was the dominant forcing in both the tropical and extratropical phases of Ophelia. Diabatic heating has also been shown more generally to be important in the development of warm-seclusion ETCs (Grønås, 1995).

**Limitations and Directions for Future Research**

Despite the improvements in dynamical models over the last few decades, it is still a challenge to accurately model TCs, even with high-resolution models. We see this in how the ECMWF Operational dataset, which is approximately the same resolution as RACMO, could not adequately simulate Ophelia's intensity, especially in the tropical phase (Fig. 2). Pairing this with the conclusion from Majumdar et al. (2023) that even the ECMWF IFS 4 km resolution cannot simulate the strength of TCs indicates that the processes present in the storm are still more complex than covered by the current model physics, especially

the hydrostatic scheme. The difference in modelling accuracy between the tropical and the extratropical phases of Ophelia may be related to the storm radius increasing and the sharp gradients decreasing as Ophelia underwent ETT. However, the examination of the reasons for differing model performance was outside the scope of this study to examine and we leave it for future research.

The tropical phase of Ophelia is more strongly affected by the change in temperature than the extratropical phase, as seen in
the sensitivity analysis in Appendix A. This indicates that the accurate modeling of PTCs is dependent on the accurate modeling of their preceding TCs. Therefore, models that do not adequately capture TCs cannot be relied on to accurately model PTCs. This reveals our need for high-resolution models that can simulate TCs *and* their transition to PTCs, as treating PTCs as pure ETCs misses part of the complexity of the system.

There is a robust signal throughout our simulations, bolstered by the sensitivity analysis in Appendix A, of storms in warmer
simulations becoming stronger and larger, maintaining their tropical characteristics longer, and becoming more of a hybrid TC-ETC storm. We expect that if this scenario were to be downscaled even further (for example with the 4km HCLIM38-AROME model; Belušić et al. (2020)) we would see a similar signal. Running such simulations were outside the scope of this study, though we believe it would allow us to more closely examine ETT and the dynamics at play that the current resolution is still too coarse to capture.

The use of uniform warming ($\Delta$T) over pseudo-global warming (PGW) is to simplify the scenario and provide a clear signal of the response to warming. While the use of a uniform warming scenario is generally fine for RACMO, with a larger domain such as the one used here the question arises if this is physically realistic. Additionally, the use of RACMO or any similar regional atmosphere model to model TCs which have such a strong coupling with the ocean water beneath it can be

questioned, as these do not capture effects like cold wake formation/cold water upwelling around the hurricane and thus allows continuous strengthening which is not possible in reality.

We know from other studies that the polar regions have been warming and will continue to warm up more than the equatorial regions (Manabe and Stouffer, 1980; Serreze et al., 2009). This could counteract some of the strengthening of the jet stream we see in our simulations due to a decrease in the meridional temperature gradient (Barnes and Polvani, 2013; Stendel et al., 2021). However, the scenario simplicity and signal clarity that the use of uniform warming introduces was determined to be worth the simplification of realistic processes for this first study into the effect of future warming on Hurricane Ophelia. Additionally, it provides two strong motivations for a future comparison with Ophelia in a PGW-based scenario: first, to determine if the near-linear increases in storm intensity and track placement we see are comparable to the results of the more complex PGW scenarios, and second to see if the altered jet stream changes Ophelia's track and impacts on Ireland.

Following the work of Evans and Hart (2003), we defined the start of ETT as the thermal asymmetry increasing over a 10 m threshold. However, this calculation is based on the height difference between two given layers of the atmosphere. With a warming atmosphere we can expect to see the height difference increase, at which point the value of 10 m would not be the same relative difference as we see now. This provides an avenue of future research, perhaps through method of long climate simulations to examine how ETT changes in a warmer world, utilizing for example the empirical methodologies of Evans and Hart (2003) or the K-clustering of Studholme et al. (2015).

We examined the impacts of Ophelia largely in terms of wind speed, which is one of the hazards a PTC brings. However, precipitation is also a significant hazard when such storms make landfall. The precipitation in a PTC often shifts to one side of the track and poleward of the storm centre, which as a result changes the impacts of the storm (Evans et al., 2017). Additionally, due to the size increase while undergoing ETT, PTCs can impact a much larger area with this precipitation. As such, precipitation is also an important factor to consider for PTC impacts, but it was outside the scope of this study because of the aforemention shift; in this case Ophelia's precipitation was shifted to the left of the track (i.e. over the ocean) and so was not substantial in terms of impacts. As such we leave it as a direction for future research.

## 6 Conclusions

Post-tropical cyclones (PTCs), the result of tropical cyclones (TCs) undergoing extratropical transition (ETT), can pose a significant hazard when they make landfall in regions unaccustomed to these powerful storms. Under future warming, it is projected that such PTCs will more often affect Europe (Haarsma et al., 2013; Baatsen et al., 2015; Sainsbury et al., 2022). To assess how such a PTC would behave under alternate climate scenarios (seven simulations, -2°C to +4°C with steps of 1°C), we modeled 2017's Hurricane Ophelia, a powerful PTC that affected Ireland, and analyzed its storm characteristics.

Our results show that in warmer climates, Ophelia becomes a larger, more intense storm, which impacted Ireland with higher winds with when passing by off the coast than in the control simulation where it hit Ireland directly. In cooler climates, Ophelia remained a weaker storm that tracked closer to mainland Europe.

The ETT of Ophelia is also different under different temperature conditions; in warmer scenarios, Ophelia becomes less asymmetrical and does so later than in cooler scenarios. Ophelia also maintains its warm core nature far longer than in the cooler scenarios. Essentially, in the warmest scenarios Ophelia does not complete ETT but remains a hybrid of a tropical and an extratropical storm.

Applying the temperature change to four more simulation initialization times showed that the signal is robust across these different initialization times: warmer simulations have stronger, larger storm, with the potential to do much more damage. Additionally, these simulations show the difference in response of the tropical and extratropical phases of the same storm to potential climate change.

The results of this study should be viewed in the light that this is a simplified case study of one storm and so cannot provide a 495 generalized answer on the impact of future TCs and PTCs on Europe. However, our results are consistent with previous studies on this topic, and this growing body of evidence indicates that there is an increased likelihood of greater impacts in warmer climates. More detailed studies have to be done of different cases, potentially with pseudo-global warming scenarios, before more definite conclusions can be drawn.

## Appendix A:  Appendix A: Sensitivity Analysis

We selected 12 October 2017 00 UTC as the initialization time due to its similarity to the observations. To examine if the conclusions we pull from the alternate climate scenarios are robust, we perform the same analyses on alternate climate simulations initialized on 11 October through 15 October at 00 UTC (see Fig. A1). However, we end the simulations at 16 October 23 UTC as insights beyond landfall in Ireland are outside the scope of this study.

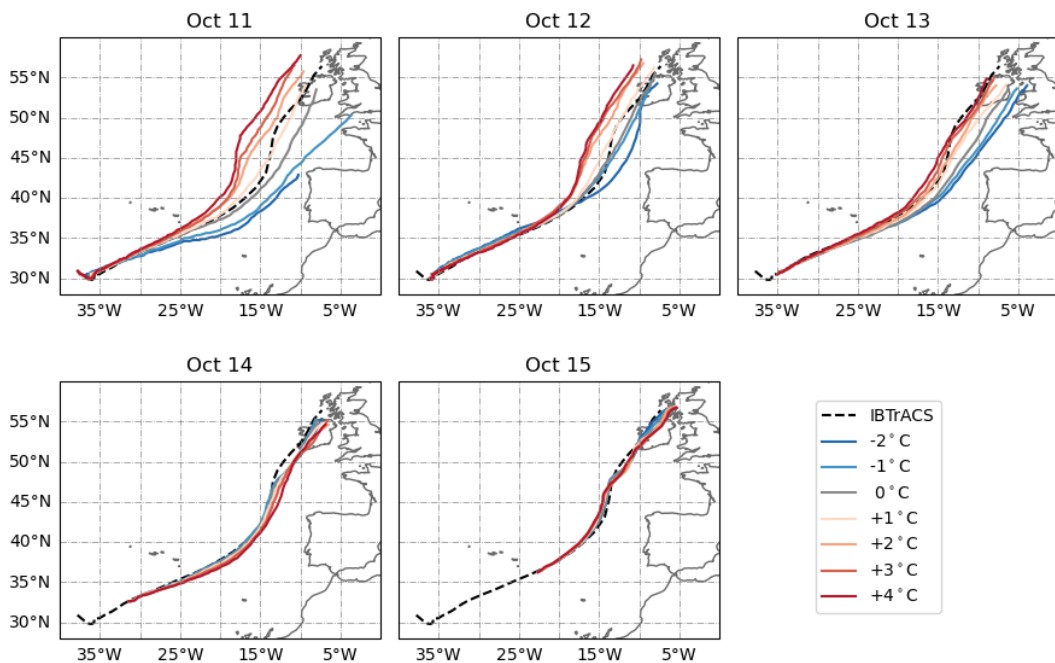

**Figure A1.** Track of Hurricane Ophelia for five initialization times (11-15 October 00 UTC) and seven temperature forcings (-2°C to +4°C). All simulations have an end time of 16 October 23 UTC. IBTrACS shows the observed track.

The tracks of the 11 and 13 October simulations follow the same pattern of shifting further westward in simulations with
higher temperature forcing. This is not the case for the 14 and 15 October simulations, which do not exhibit a clear pattern but generally all cluster in one group. The spread of the tracks shows a substantial decrease with later initializations, clear difference between 14 and 15 October and the simulations initialized before then.

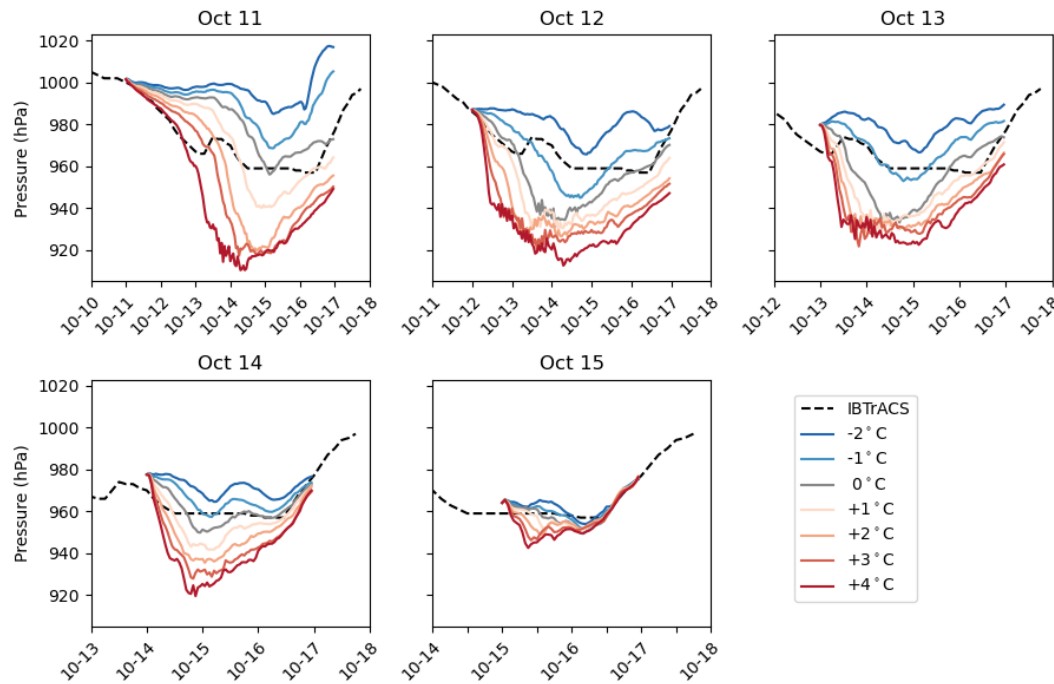

**Figure A2.** Minimum pressure profiles of Hurricane Ophelia for five initialization times (11-15 October 00 UTC) and seven temperature forcings (-2°C - +4°C). IBTrACS shows the observed pressure.

Similarly, we can examine the time series of central pressure for the simulations with different start dates. Figure A2 shows a consistent trend across initialization dates that in simulations with higher temperature forcing Ophelia has a lower central pressure. These lower central pressures are also accompanied with greater intensification rates. The spread of minimum central pressures across different temperature forcings is less with later simulation initialization time: 11 October has a difference of 78 hPa where 15 October differs by only 11 hPa, and we see a much higher minimum central pressure on 11 October than the other initialization dates. This suggests that the effect of the alternate climate on the 15 October run is less. This could be due to less time under conditions favourable for strengthening or because the nature of the storm has changed to one less affected by changing climate.

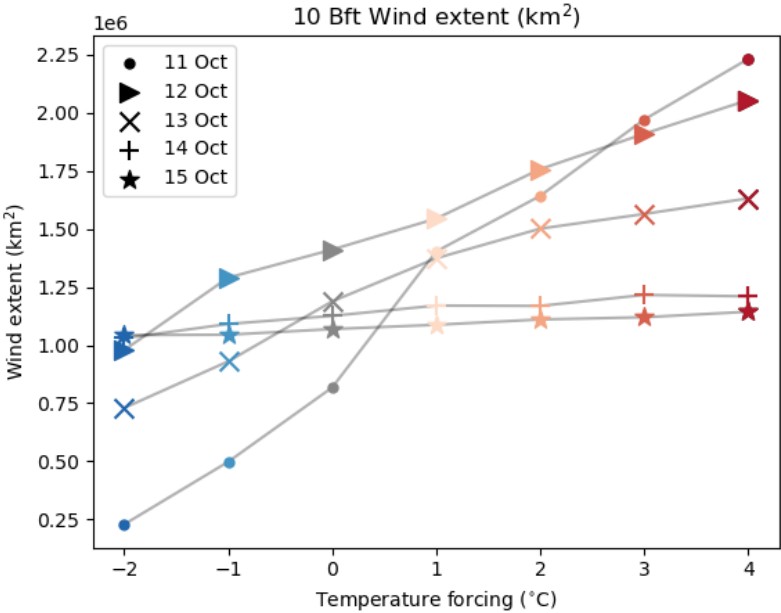

**Figure A3.** Accumulated 10 Bft wind extent (km$^2$) after 15 Oct 00 UTC for each set of simulations, plotted against temperature forcing.

In section 4.2 we saw that increased temperature forcing led to a greater extent of high winds. We performed the same analyses on the other initialization times and plotted the results in Fig. A3. To prevent the bias of longer simulations automatically having larger wind extents, we select only the data present in the timespan shared by all the simulations, 15 October 00 UTC to 16 October 23 UTC.

For each simulation, we observe a positive relationship between increased temperature forcing and increased wind extent. This is consistent with the findings of (Radu et al., 2014), who note that under warmer climate conditions the TC radius increases. This effect is most pronounced in the 11 October simulations, which show an increase of 877% in the wind field between the -2°C and +4°C simulations. 12 and 13 October show modest 110% and 124% increases in wind field extent, while the 14 and 15 October simulations, in contrast, change very little with increasing temperature forcing (17% and 10% increases respectively). This suggests both that the 14 and 15 October simulations do not have enough time to adjust to the new climate conditions before undergoing ETT, and that ETC size is less affected by changes in climate conditions than TC size.

When completing the same storm size analysis as in Figure 4(c), we come to the same conclusion (see Figure A4). The variation in size is much more pronounced in the 11 October simulations than for the 14 or 15 October simulations, with the storms in the warmer simulations having a larger diameter than those in cooler simulations. Figure A5 shows the same data as FigureA4, however grouped by $\Delta$T rather than initialization time. We see that for the same level of $\Delta$T, storms that are initialized earlier have a larger storm size than those initialized later. This is likely because earlier initializations have more time to both adjust to the alternate climate conditions.

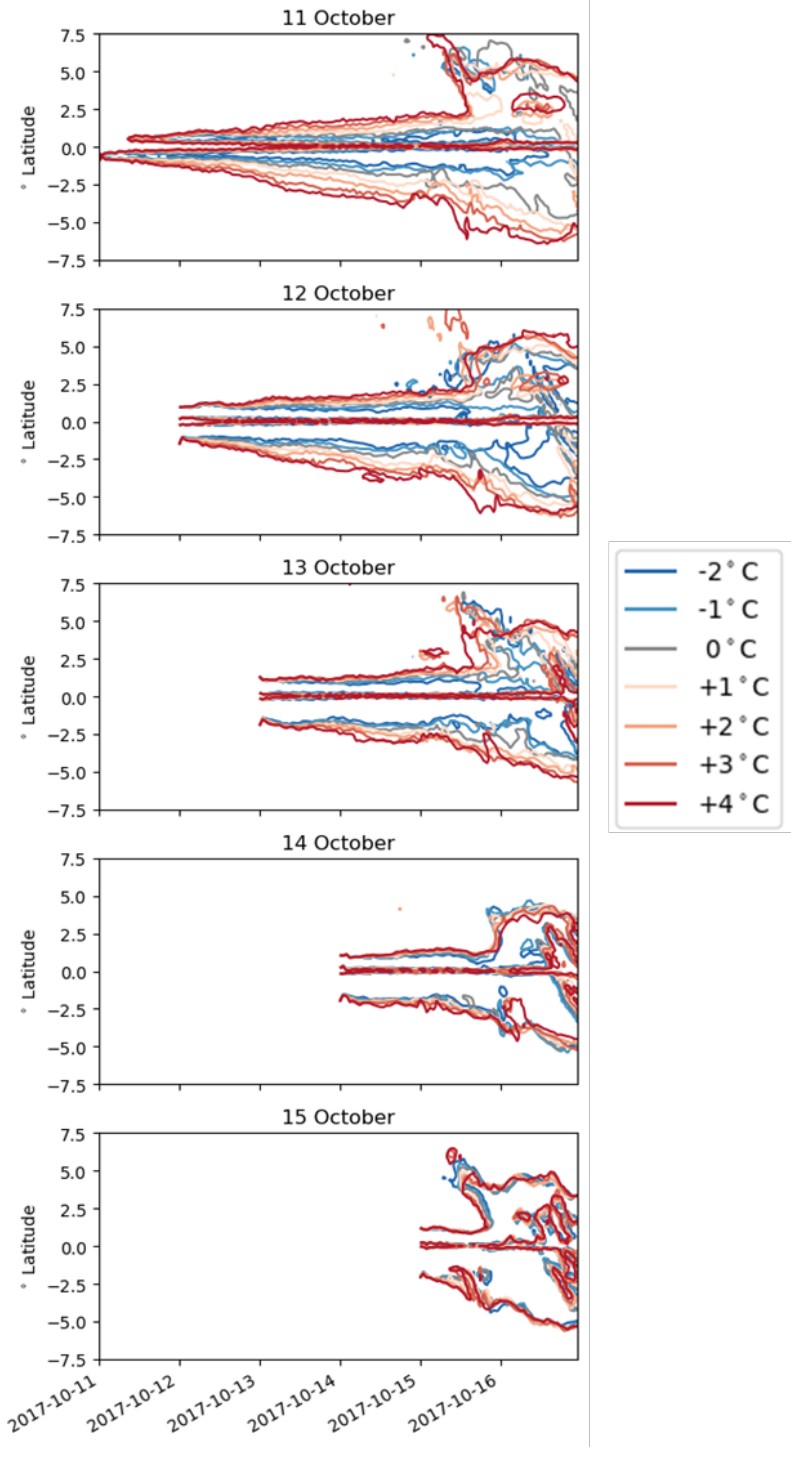

**Figure A4.** North-south diametric slice of extent of 10m instantaneous 17 m/s wind, in degrees relative to storm centre for each of the 5 simulation times and 7 ΔT levels, grouped by simulation time.

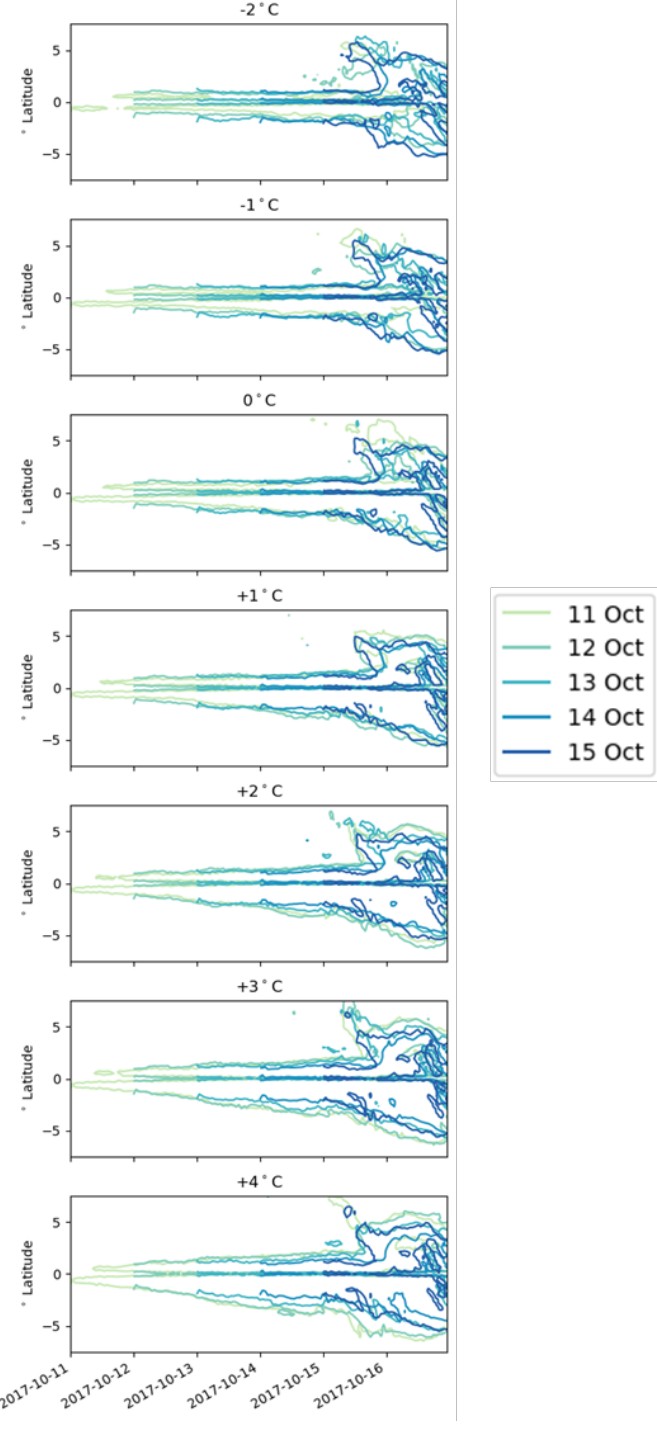

**Figure A5.** North-south diametric slice of extent of 10m instantaneous 17 m/s wind, in degrees relative to storm centre for each of the 5 simulation times and 7 ΔT levels, grouped by ΔT level.

We also examine the variation of the beta drift between the different simulation levels and initialization times. Both $R_{max}$ and $V_{max}$ are included in the calculation of the beta drift, and we have plotted these along with the beta drift in Figure A6. The

535 $R_{max}$ values tend to decrease at the beginning of the simulation, with the warmer simulations showing a much sharper decrease than the cooler simulations (Figure A6, left column). This can also be seen in the contraction of the eye in the storm size plots of Figure A4. At the same time we see an increase in maximum wind speed (Figure A6, middle column). All of these point to a strengthening of the storms due to a conservation of angular momentum. Beta drift, as a combination of these two factors, shows an initial decrease followed by a gradual increase and general stratification by applied $\Delta$T (Figure A6, right column).

However, based on the beta drift equation, we expect that $R_{max}$ has a greater effect on the beta drift than $V_{max}$. This may be a contributing factor to the relative lack of difference between beta drift values across the 11 and 12 October simulations despite larger differences in $V_{max}$ (see Appendix B3 for the derivation). In the 14 and 15 October simulations, the difference in $R_{max}$ between the simulations is not as pronounced. Additionally, there is not as much time for the slow expansion of the $R_{max}$ after the initial decrease due to strengthening and before the rapid increase also visible in the storm size plots (Figure A4).

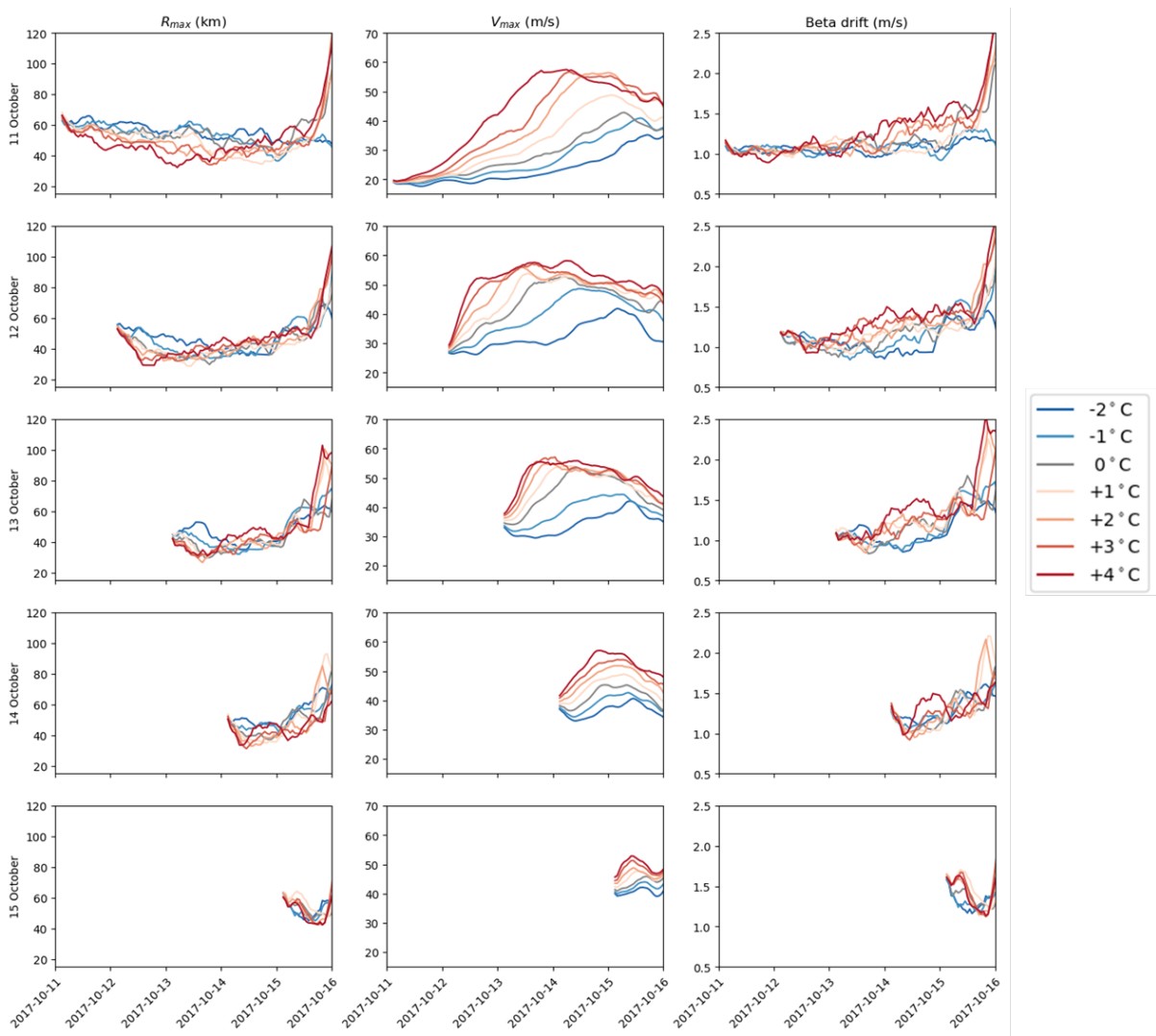

**Figure A6.** $R_{max}, V_{max}$ and beta drift for the 5 simulation dates (11-15 October) and 7 $\Delta T$ levels (-2 to +4). 6 hour convolution applied to all variables, so plotting starts at 6 hours past initialization time as labeled. Time axis capped at 2017-10-16 to examine tropical phase.

## 545  Appendix B: Appendix B

### B1  RACMO Specifications

RACMO was run with 40 hybrid sigma levels, as defined by:

$$\eta_k = A_k + B_k * p_{surf}, (k = 1 - 40)$$

$A_k$ is the pressure coeficient and $B_k$ is the sigma coefficient. $A_k$ and $B_k$ are chosen so that B dominates in the bottom of the troposphere and A dominates above.

**Table B1.** Greenhouse gas concentrations in RACMO simulations, as taken from CMIP5-RCP8.5 for October of 2017

| Gas | $CO_2$ | $CH_4$ | $N_2O$ | $CFC11$ | $CFC12$ |
|---|---|---|---|---|---|
| Concentration | 407.922 | 1.87473 | 329.009 | 210.227 | 495.062 |
| | ppmv | ppmv | ppbv | pptv | pptv |

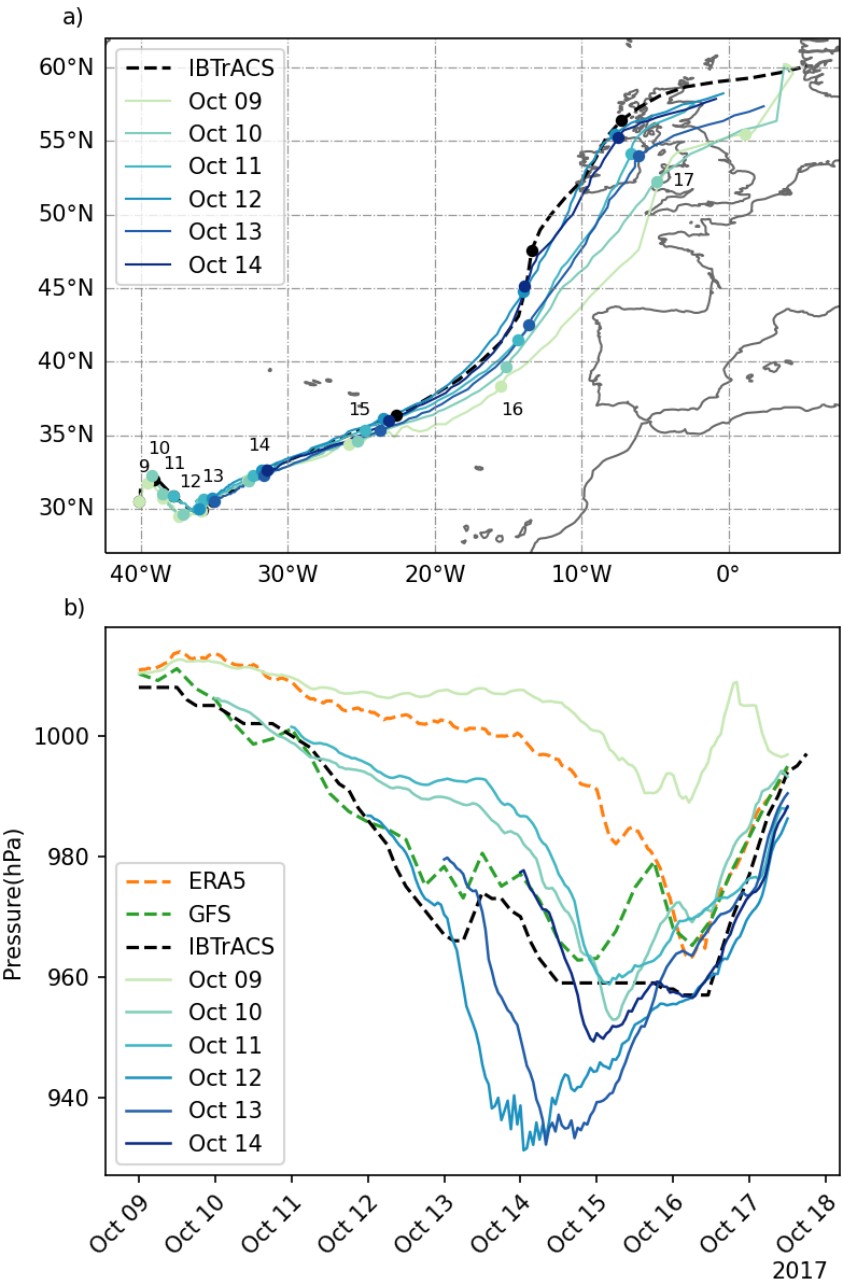

**Figure B1.** Track (a) and minimum central pressure (b) for Hurricane Ophelia for the current climate downscaled RACMO simulations with GFS boundaries with varying initialization times. The black dashed line is the IBTrACS observations, and the pressure time series derived from the ERA5 reanalysis and the GFS analysis data are included in (b).

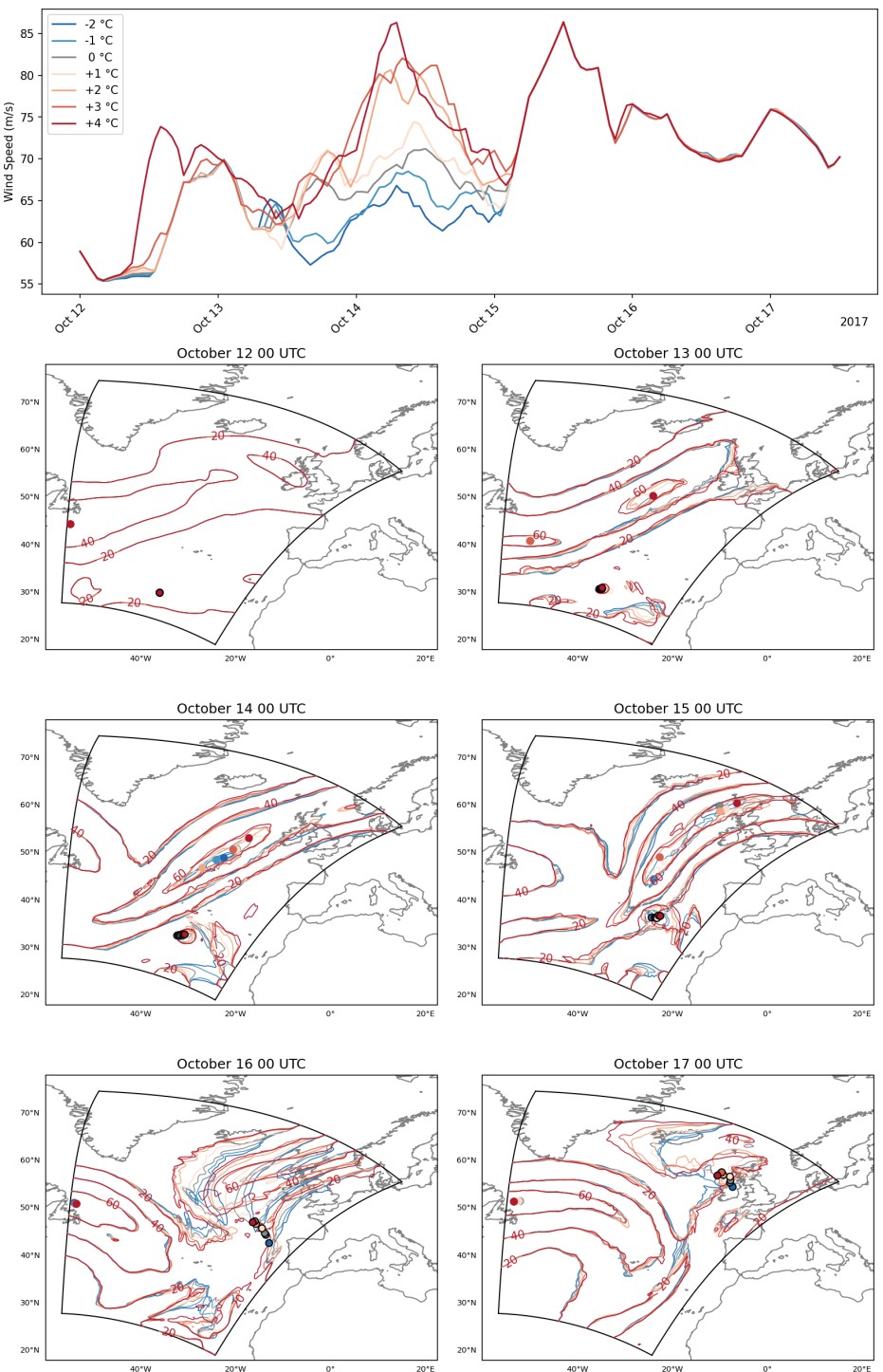

**Figure B2.** 200 hPa wind speed maximum (top) and contours at 12 - 17 October 00 UTC for each temperature scenario, 20 m/s contours, with the position of Ophelia at each time plotted as an outlined circle, with location of maximum wind as an unoutlined circle. Polygon indicates edges of RACMO domain, as in Fig. 9.

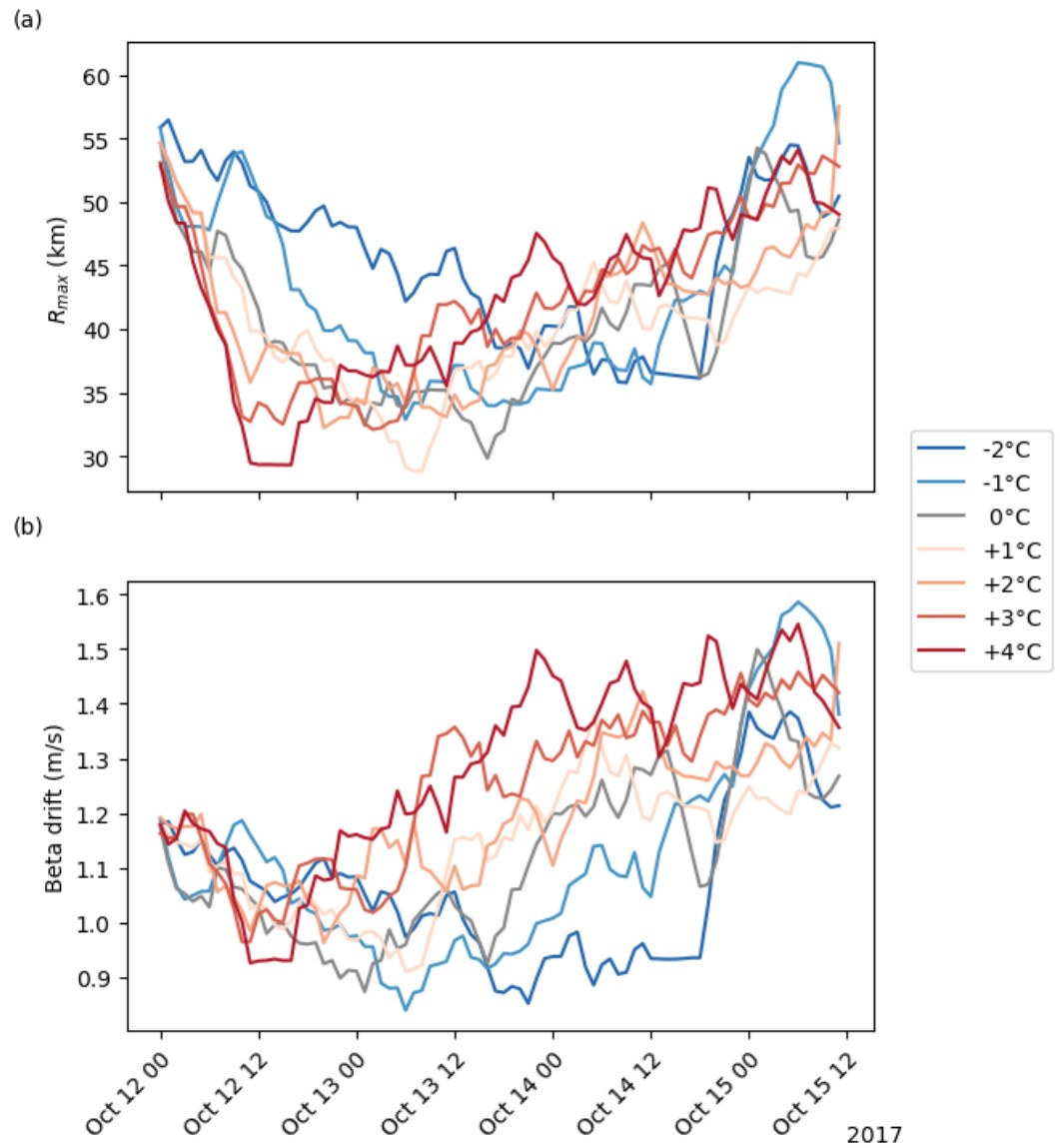

**Figure B3.** (a) Radius of max wind speed ($R_{max}$) in km and (b) beta drift in m/s for Ophelia in each alternate climate scenario. Beta drift calculated following the hurricane beta drift law from Smith (1993)

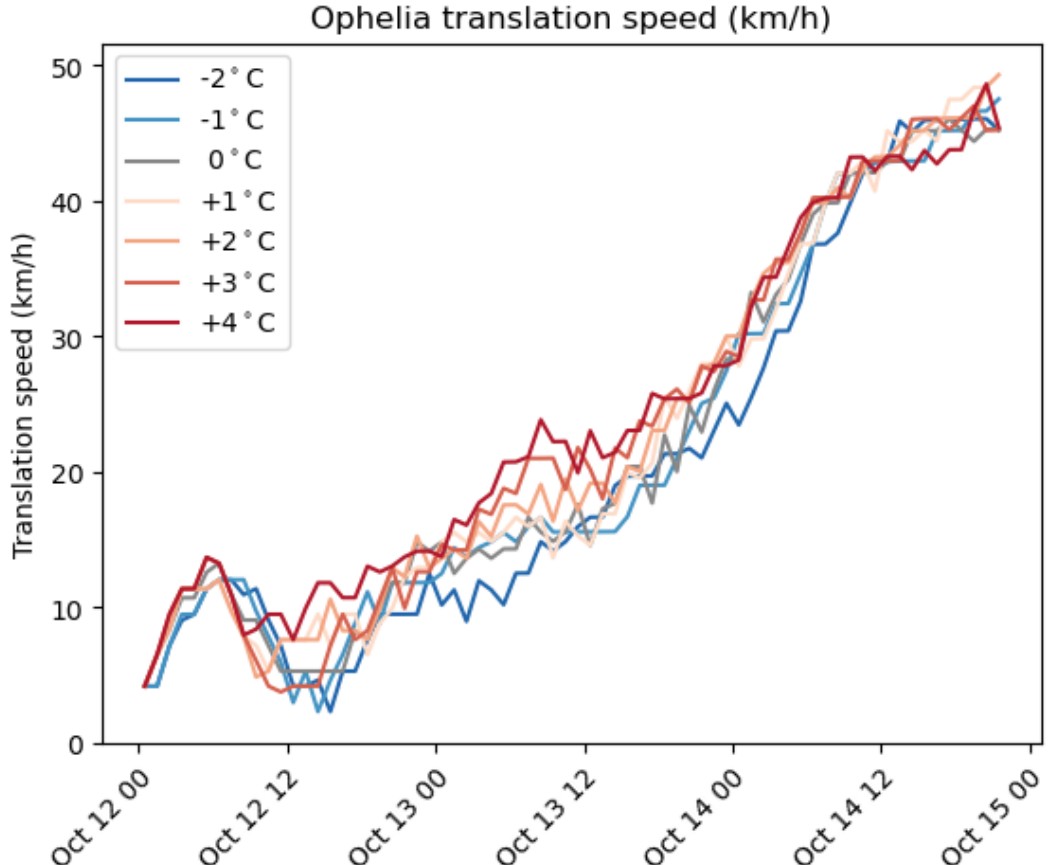

**Figure B4.** Translation speed of Ophelia in km/h for each scenario for the first 72 hours of the simulations.

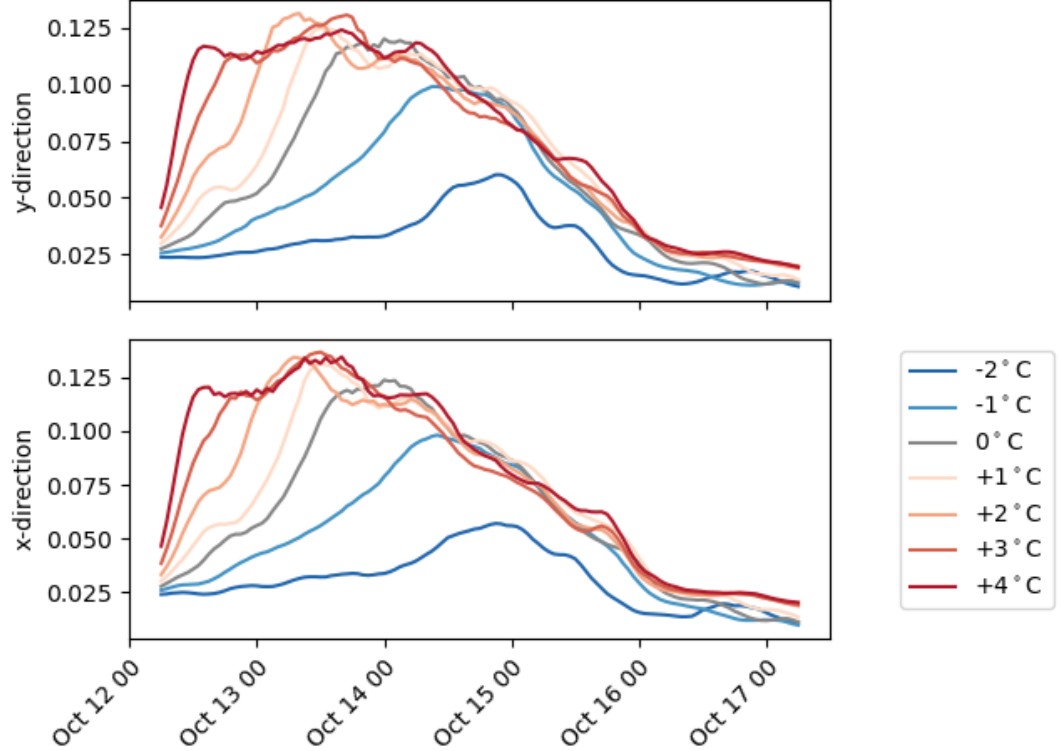

**Figure B5.** Maximum horizontal and vertical pressure gradient values around Ophelia for each simulation. A 6-hour convolution was applied.

.

## B3    Beta drift derivation

The beta drift formula as taken from Smith (1993):

$$BD = 0.72B^{-0.54}r_{max}^2\beta$$

where

$$B = \frac{r_{max}^2\beta}{V_{max}}$$

$$BD = 0.72\left(\frac{r_{max}^2\beta}{V_{max}}\right)^{-0.54}r_{max}^2\beta$$

$$BD = 0.72\left(\frac{V_{max}}{r_{max}^2\beta}\right)^{0.54}r_{max}^2\beta$$

$$BD = 0.72(V_{max})^{0.54}(r_{max}^2\beta)^{0.46}$$

$$BD = 0.72\,V_{max}^{\,0.54}\,r_{max}^{0.92}\,\beta^{0.46}$$

*Author contributions.* MR and HdV planned the experiments, and EvM carried out the simulations. MR, HdV, NB, and MB interpreted the results. MR prepared the manuscript with contributions from all coauthors.

*Competing interests.* The authors declare no competing interests present.

*Acknowledgements.* We acknowledge the use of imagery from the NASA Worldview application (https://worldview.earthdata.nasa.gov), part of the NASA Earth Science Data and Information System (ESDIS). Thanks go to John Hanley and Tatjana Kokina for their contributions to fruitful discussions.

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
