# Peer review of "Tropical cyclone intensification and extratropical transition under alternate climate conditions: a case study of Hurricane Ophelia (2017)"

_EGUsphere, 2025_

## Referee Comment (RC1)

**Review of WCD-2025-218**
"Tropical cyclone intensification and extratropical transition under alternate climate conditions: a case study of Hurricane Ophelia (2017)"

by
Marjolein Ribberink, Hylke de Vries, Nadia Bloemendaal, Michiel Baatsen, and Erik van Meijgaard

**Recommendation: Moderate Revisions**

Ribberink et al. (2025) investigates the effect of warmer temperatures on the structure and evolution of Hurricane Ophelia (2017). The authors find that warmer temperatures led to a stronger storm with larger wind field. Additionally, Ophelia's ETT was delayed and the storm maintained TC characteristics longer in the warmer environments.

This study is well-motivated by the growing concern of changes in ETT in the future and the likely impact on Western Europe. The manuscript is very well written and easy to follow. The methods all seem reasonable and consistent with previous work, and I appreciate including the dataset comparison to justify your choice of model initial conditions. The results are also in line with previous work on ETT and climate change.

I would, therefore, rate the overall presentation quality of this manuscript as "excellent", the scientific significance as "excellent-to-good", and the scientific quality as "excellent-to-good". I have just a few comments and clarification questions before recommending publication in *Weather and Climate Dynamics*.

General Comments

(1) Here are some additional ETT references to consider for inclusion throughout the introduction:

Arnott et al. (2004): https://doi.org/10.1175/MWR2836.1
Bieli et al. (2019): https://doi.org/10.1175/JCLI-D-17-0518.1
Bieli et al. (2020): https://doi.org/10.1029/2019MS001878
Kofron et al. (2010): https://doi.org/10.1175/2010MWR3180.1
Wood and Ritchie (2014): https://doi.org/10.1175/JCLI-D-13-00645.1

(2) The following references may be useful additions to the discussion in L411–416:

Bieli et al. (2020): https://doi.org/10.1029/2019MS001878
Zarzycki et al. (2017): https://doi.org/10.1002/2016MS000775

Specific Comments

L11: Change "Post" to "post".

L80: It looks like there may be a missing equation for the wind speed conversion?

L102: Replace "spatial resolution" with "horizontal grid spacing" as the two terms are not synonymous.

L113: Should introduce PGW acronym here since it's used later.

L113: As a side note, PGW can also just involve altering thermodynamics rather than altering dynamical fields. The difference is that the temperature deltas are horizontally and vertically variable rather than uniform (e.g., Lackmann 2013; Lackmann 2015; Jung and Lackmann 2019).

L115: I appreciate taking a simple, constrained approach to this experiment, but wouldn't the effects of a north/south jet shift, if it occurred, also be a meaningful result?

L169: How many pressure levels and at what spacing did you use for your simulations?

L179: What's the shorter interval?

L196: What does 12 October indicate? Is this needed in the heading title?

L203–204: This is an interesting result—can you put this into context with previous work suggesting a potential TC slow down?

Figure 4: Panels (b)–(d) are very difficult to interpret due to the noise. Suggest applying a smoother to help highlight the signal.

L234–235: Can you clarify this statement about reaching the model's capability?

Figure 5: I believe the legend is missing here.

L300: Can you elaborate on the increase in symmetry towards the end of Ophelia's lifecycle? It this pointing to the storm reintensifying as an ETC and undergoing warm seclusion?

Table B1: I don't see Table B1 referenced anywhere in the text. This does raise a question though—was $CO_2$ increased in the future simulations?

---

## Author Response (AR1)

**Tropical cyclone intensification and extratropical transition under alternate climate conditions: a case study of Hurricane Ophelia (2017)**

**Reviewer 1**

**Ribberink et al. (2025) investigates the effect of warmer temperatures on the structure and evolution of Hurricane Ophelia (2017). The authors find that warmer temperatures led to a stronger storm with larger wind field. Additionally, Ophelia's ETT was delayed and the storm maintained TC characteristics longer in the warmer environments.**

**This study is well-motivated by the growing concern of changes in ETT in the future and the likely impact on Western Europe. The manuscript is very well written and easy to follow. The methods all seem reasonable and consistent with previous work, and I appreciate including the dataset comparison to justify your choice of model initial conditions. The results are also in line with previous work on ETT and climate change.**

**I would, therefore, rate the overall presentation quality of this manuscript as "excellent", the scientific significance as "excellent-to-good", and the scientific quality as "excellent-to-good". I have just a few comments and clarification questions before recommending publication in Weather and Climate Dynamics.**

We thank the reviewer for investing their time in reading this paper, as well as their kind words and positive evaluation of our manuscript.

**General Comments**

(1) **Here are some additional ETT references to consider for inclusion throughout the introduction.**
(2) **The following references may be useful additions to the discussion in L411–416**

We thank the reviewer for the suggestions and have incorporated these papers into the manuscript where they contributed to the discussion.

**Specific Comments**

**L11: Change "Post" to "post"**

Done.

**L80: It looks like there may be a missing equation for the wind speed conversion?**

The authors thank the reviewer for pointing this out and have included the equation. This will be adjusted to:

*"All wind speed measurements were converted to 10-minute 10-meter maximum sustained wind speed, following Harper et al., (2010):*

$$V_{600} = 0.93V_{60} \ (1)\text{"}$$

**L102: Replace "spatial resolution" with "horizontal grid spacing" as the two terms are not synonymous.**

Done. We have also replaced this where it occurred elsewhere in the text (Lines XX (analysis data)

**L113: Should introduce PGW acronym here since it's used later. As a side note, PGW can also just involve altering thermodynamics rather than altering dynamical fields. The difference is that the temperature deltas are horizontally and vertically variable rather than uniform (e.g., Lackmann 2013; Lackmann 2015; Jung and Lackmann 2019).**

Following a comment made by Reviewer 2, we have decided to omit the term PGW altogether, and have changed this to "alternate climate scenarios".

**L115: I appreciate taking a simple, constrained approach to this experiment, but wouldn't the effects of a north/south jet shift, if it occurred, also be a meaningful result?**

This is a fair point raised by the reviewer. If such a meridional jet shift occurred, as indicated in recent studies (Rivière, 2011; Woollings et al., 2023; Yin, 2005) it would indeed be a meaningful result. However, a northward shift of the jet stream without an accompanying shift by Ophelia would run the risk of the storm not being picked up by the jet stream. While this is indeed a plausible result, the goal of this study was to look at how Ophelia's evolution and extratropical transition would be affected by climate change. If the storm does not or barely undergoes extratropical transition in warmer scenarios, we end up modelling a different scenario than a storm that does undergo transition and becomes a powerful PTC that can then bring large impacts. We see from

a study such as that by Ritchie & Elsberry (2007) that there can be large differences in storm track and storm strength dependent on initial TC-jet stream phasing and placement.

We do know from other studies that storm tracks will also shift poleward in concert with the jet shift (Anjana & Kumar, 2023; Studholme et al., 2022; Tamarin-Brodsky & Kaspi, 2017). While we could have shifted Ophelia poleward following outcomes of these studies to prevent this change in interaction pattern, this would introduce other uncertainties to our results. Especially because Ophelia was already an edge case that the model had trouble picking up and strengthening when not starting with a significant storm already in the basin (see Figure B1) changes like this are likely to have a massive impact on if the storm can be meaningfully modelled at all.

To summarize, while such a jet stream shift would be a meaningful result, it would give us much greater uncertainties and a much larger spread in possible outcomes that it would be difficult to draw any meaningful conclusions about what the effect of climate change is on Hurricane Ophelia specifically.

We will be adding a discussion on this to the section on RACMO Description, see lines 133-137:

*In an attempt to minimize displacement of the systems, we apply a spatially uniform ΔT to current climate forcing of both the atmosphere and the sea surface temperature. This prevents temperature gradients that could shift the jet north or south of its observed position. While this is an approximation, it is one that is included in order to minimize the spread of possible outcomes such as the jet stream never interacting with Ophelia and its ETT never occurring at all. Our aim is to be able to draw conclusions on how climate change would affect specifically Ophelia.*

**L169: How many pressure levels and at what spacing did you use for your simulations?**

The pressure levels for output were limited, but the model itself has far more hybrid sigma levels. The model itself has 40 hybrid levels, as defined by:

$$\eta_k = A_k + B_k * p_{surf}$$

where $k = 1 - 40$ and $A_k$ is the pressure coefficient and $B_k$ is

the sigma coordinate ($p/p_{surf}$).

$A_k$ and $B_k$ are chosen so that B dominates in the bottom of the troposphere and A dominates above.

We will be adding this information to the section on RACMO description and Appendix B, see lines 115-119:

 *"RACMO is an uncoupled hydrostatic atmosphere model with a spatial resolution of 12 x 12 km, with 40 hybrid sigma levels, and model physics based on ECMWF IFS Cy33r1 but an adjusted boundary layer scheme... More detailed specifications can be found in Appendix B and Table B1."*

**L179: What's the shorter interval?**

The shorter interval is the 12-hour smoothing we use rather than 24-hour smoothing used in Hart, (2003). This is explained in Section 3.5, see lines 200-204:

*"The calculated $B$, $-V_T^U$ and $-V_T^L$ data are smoothed using a 12-hour convolution filter to remove noise present due to the chaotic nature of the track, where slight shifts can change the angle of propagation and thus the area over which these variables are calculated. Hart (2003) used a 24-hour running mean filter for the same purpose, however due to the higher temporal resolution of our simulations (hourly vs their 6-hourly) we use a shorter interval."*

**L196: What does 12 October indicate? Is this needed in the heading title?**

12 October is the initialization time of the simulation we study. The date was added to distinguish it from the other start times analysed later (11-15 October; see Appendix A). However, we understand this might raise confusion amongst our readers, and, as this was discussed at length in Section 2.3, we have decided to remove it from the heading title and replace it with just *Alternate Climate Scenarios*.

**L203–204: This is an interesting result—can you put this into context with previous work suggesting a potential TC slow down?**

Over the last several years there has been a growing interest in how the TC translation speed may change. Kossin (2018) studied TCs that occurred between 1949-2016, and found a ten percent decrease in global translation speed. However, when splitting by basin and separating on-land vs at-sea, Kossin (2018) found also that the Atlantic had a much smaller translation speed decrease at-sea, as Ophelia was at this time, than on-land (-6 % vs -16%).

There is however a discussion on whether the changes seen at this time and reported by Kossin (2018) are truly changes seen in TC slowdown or if these are data artefacts resulting from changes in measurement practices and the incorporation of satellite data in the 1970s (Chan, 2019; Lanzante, 2019; Moon et al., 2019). When excluding these earlier times, Sun et al. (2021) actually find a positive trend in the North Atlantic TC

translation speed in the period of 1982-2016. D. Zhang et al., (2020) also found a decrease in translation speed in the North Pacific in the period 1949-2017, but with no significant trend after 1981. Yamaguchi et al., (2020) ran simulations of both historical and future (+4K) climates, and find no slowdown in the period 1951-2011, and even find an increase in translation speed in the period 2051-2110, though much of this is attributed to a poleward shift in cyclone tracks, as translation speeds tend to be greater at high latitudes. Zhang et al. (2020) describe that while TC slowdown will be greater in the midlatitudes, this is largely attributable to a poleward shift in the midlatitude westerlies. Since our wind patterns do not see this poleward shift, we also do not expect to see a substantial slowdown.

Another interesting conclusion from Sun et al. (2021) is that stronger TCs tend to be controlled more by upper level flow, and weaker TCs tend to be controlled by more lower level flow. We find that our simulated jet stream is stronger in the warmer scenarios, where the storm is also stronger, these two things likely compound and result in a higher translation speed for the stronger storms.

We have added sections of this discussion to the manuscript, see lines 222-223 and 311-314.

**Figure 4: Panels (b)–(d) are very difficult to interpret due to the noise. Suggest applying a smoother to help highlight the signal.**

Following the reviewer's suggestion, we have applied a 6-hour running mean to the data for panels (b)-(d) in Figure 4.

**L234–235: Can you clarify this statement about reaching the model's capability?**

Due to the complex processes that take place in the cores of TCs, small horizontal grid spacing is required to be able to accurately model the storms with any degree of accuracy. However with high spatial resolution comes the requirement to also decrease the timestep, to prevent the model from approaching or surpassing the CFL criterion.

This can be seen in the $R_{max}$ (Radius to maximum winds) values dropping to the order of 30-35 km or 2-3 grid cells, which equates to an eye covered by no more than 4-6 grid cells in either direction (Figure B3(a)). With the complicated system of updrafts, subsidence, and other sharp gradients present within the eye, it is not possible to fully model this, and thus parametrization needs to be done by the model. This is why we have also included this as a potential direction for future research is to run this with a finer-resolution model to better resolve the dynamics. We have added sections of this discussion to the manuscript, see lines 259-262.

**Figure 5: I believe the legend is missing here.**

We thank the reviewer for pointing this out. We have added a legend to Figure 5.

**L300: Can you elaborate on the increase in symmetry towards the end of Ophelia's lifecycle? It this pointing to the storm reintensifying as an ETC and undergoing warm seclusion?**

Yes, this is indeed a signal of the storm reintensifying and undergoing warm seclusion. We see this in the typical warm seclusion lifecycle (Dekker et al., 2018; Hart et al., 2006; Sarro & Evans, 2022). Sarro & Evans, (2022) actually identifies Ophelia as an "instant warm seclusion", similar to "seclusion-occlusion" events identified by Kitabatake (2008), where TCs immediately have a warm-core structure after interactions with a strong upper-level trough.  See also the newly-added Figure 6, which clearly shows the progression from symmetric TC to asymmetric transitioning cyclone to symmetric PTC.

We have added this information to the manuscript, see lines 335-343:

*"All simulations except the -2◦C have a peak in asymmetry at the end of 15 October, after which their 300 asymmetry decreases again. The three warmest simulations very quickly attain low levels of asymmetry and return to near-TC levels. These results are indicative of an "instant warm seclusion", a term coined by Sarro & Evans (2022), to describe a storm that immediately attains a storm structure similar to that of a warm seclusion cyclone but without going through the cold-core structure typical of that pathway. As the warmer scenarios return to TC-like faster and with less asymmetry than the cooler simulations, this implies that Ophelia maintains more of its TC-like characteristics in the warmer scenarios. This progression can also be seen in Figure 6: while all scenarios undergo the transition from radially symmetric TC on 14 October to asymmetrical on 16 October, we see that by 16 October 12 UTC the storms in the warmest scenarios return to the radially symmetric structure more than the cooler simulations."*

**Table B1: I don't see Table B1 referenced anywhere in the text. This does raise a question though—was CO2 increased in the future simulations.**

CO2 was not increased in the future simulations; all of the simulations have the same greenhouse gas concentrations. We referred solely to Appendix B and not Table B1. We apologize for this confusion and have added a direct reference to Table B1 in Section 3.3, see lines 119.

**Tropical cyclone intensification and extratropical transition under alternate climate conditions: a case study of Hurricane Ophelia (2017)**

**Reviewer 2**

**Overall assessment: In general, I find this study both interesting and valuable to the scientific community. It provides insights into the response of post-tropical cyclones to a warmer climate and their potential impacts on regions like Western Europe. Additionally, the study explores the physical mechanisms behind changes in the storm's structure, behavior, and impact in response to various warming and cooling scenarios. However, I do have some concerns and/or required clarifications. These are highlighted in subsection below. I suggest publishable with major revisions, since I believe the authors need to make some rather substantial text modifications and additions to fine-tune the message of this study. I think the manuscript will be publishable after some more work.**

We thank the reviewer for their kind words and their time and effort taken to review our manuscript.

**Major comments:**

1) **Introduction: The introduction is generally well-written, providing the background and motivation for this study. However, the specific research question or hypothesis could be clearer. For example, explicitly stating what the paper aims to address or how it intends to fill a gap in the current literature would strengthen the introduction further. Additionally, details about the chosen case could be moved to later sections (e.g., Section 3.1; this section is too short), allowing the introduction to remain focused and concise, while the authors briefly mention it. Aside from the introduction, the following sections are quite similar, with only a few lines presented in each.**
We thank the reviewer for suggesting the clarification. We have moved the case study details into their own section and clarified the aim of the paper.
The end of the introduction has been adjusted, see lines 51-60:

*"There have been many case studies on PTCs in the last several decades, especially in the North Atlantic (Atallah and Bosart, 2003; Evans and Hart, 2008; Feser et al., 2015; Galarneau et al., 2013; Jung and Lackmann, 2019; McTaggart-Cowan et al., 2004; Thorncroft and Jones, 2000). However many of these focused on US-impacting storms, and those that do impact Europe are only studied in current climate conditions. As far*

*as the authors are aware, no specific case studies have been done on Europe-impacting transitioning storms incorporating climate change factors. In this study, we therefore aim to fill this gap by examining the changes in the structure, behaviour, and impacts of Hurricane Ophelia under alternate climate scenarios, paying particular attention to the changes in Ophelia's ETT. We use a ΔT approach utilizing the Regional Atmosphere Community Model (RACMO) model. The results show that Ophelia becomes a larger and stronger storm under warmer climate conditions. The outcomes can be used to demonstrate the increased risk posed by the expected increase in such storms under climate change conditions."*

2) **Introduction: It would be helpful to include a brief outline of the paper, as this can guide the reader and give them an idea of what to expect in each section.**

   We appreciate the suggestion, and have added such an outline to the introduction. See lines 61-62.

   *"Section 2 provides a description of the case study. The data and methods used in this paper are described in Section 3. We present the results of our analyses in Section 4 and discuss the study in Section 5. Finally we conclude the paper in Section 6."*

3) **Section 2: This section contains numerous subsections, many of which are only a few lines long. Please consider consolidating these subsections to make the content more concise. For example, sections 2.6.1 and 2.6.2 can be combined.**

   Upon recommendation of the reviewer, we have combined the subsections of 2.6 into one section, simply titled "Quantifying Extratropical Transition". Additionally, sections 2.4 and 2.5 have been combined into "Cyclone tracking and Footprint".

4) **Lines 220-222: If this is the case, I think the large deficit in MSLP (e.g., 912 hPa) should not be captured in these simulations either. There seems to be misrepresentations somewhere in the dynamical processes that prevent the storm from achieving the gradient wind balance. Additionally, if the authors used instantaneous maximum wind speeds recorded at specific time intervals (e.g., hourly output data; please also specify the frequency of output from your simulations), they might have missed the actual maximum wind speed that occurred between these intervals. On the other hand, as the authors described, IBTrACS provides 1-min average maximum sustained wind speed.**

   To the first point, while the model is limited in how realistically it represents the gradient wind balance, it still represents a storm that looks realistic in terms of wind field and overall structure. This is not uncommon for regional models, for instance Arpège can also severely overestimate TC intensity but still generally has a realistic representation

(Chauvin et al., 2020). There is clearly a limitation to the model here, but the simulations do well otherwise to represent the intensity and evolution of Ophelia.

To the second point, we have used instantaneous maximum wind (hourly output data). We have added this to the manuscript to further clarify our process (see caption of Figure 4). However, as a final note, there are no direct observations of Ophelia, only estimates based on satellite representation (Stewart, 2018). We have no way of ruling out that Ophelia at some point had an actual core pressure of 930-940hPa, albeit unlikely.

5) **Lines 223-225: After October 16, it is possible that all the simulated storms begin interacting with land, which could significantly influence wind speed. Although the +2~+4 storms are located further west compared to the cooler ones, their larger storm radius suggests they may begin interacting with land. Line 243 may support this speculation. Please consider adding information about the size of the simulated storms for clarification.**

This is a possibility, but this would mainly be the outer regions of the storm, which may not have as much effect on the peak intensity of the storm, which is present near the core of the storm, which is still far out at sea (hundreds of km away). Additionally, the 10m wind footprints in Figure 8 don't seem to support this hypothesis of land interaction. The contour plot in Figure 8 shows a westward movement of the 20 m/s contours under warmer scenarios, indicating a shift *away* from land.

A different contributing factor to the similarity in peak wind speed between may be that despite the storms in the warmer scenarios having a much deeper central pressure, they also grow in size.  We therefore believe that the pressure gradients, which directly influence the peak wind speeds, are quite similar amongst the different climate runs, thereby keeping the wind speeds approximately the same across the different runs.

To better understand this, we have plotted wind speed plots at 17 m/s contour (storm size) at the moment of first contact with land by the body of the storm* for each of the simulations as this served the same purpose as the more complicated distance to land and storm size calculation indicated earlier (see Figure R5).  What we see there is that the reviewer certainly has a point – the warmer storms generally interact with the land earlier than the cooler storms, with a range of 16 October 5 UTC to 10 UTC.

We see that the dropoff of wind speeds in Figure 4(d) does start earlier than the first contact however, generally around 16 October 00 UTC, which is interesting since the first contact is with the very outer edges of the storm. Additionally, at 16 October 00 UTC all the storms are still quite far out to sea (Figure R6). This would point to the interaction

with land being a contributing but perhaps not the main factor of the similarity in windspeeds.

When examining the pressure gradients, we find that after 16 October they are all very similar across alternate climate simulations (see Figure R7). We actually see quite a similarity in the pressure gradient plots and the wind speed plot (Figure 4(d)). These similarities would indicate that despite the differences in strength that exist between the different simulations, the greater size of the warmer storms brings the gradients and thus the winds to similar values.

We have inserted part of this discussion into the manuscript, see lines 244-248.

*Almost all storms had a "body" and a "tail" in the plots. The tail, often ragged, frequently interacted with the land earlier than the main body of the storm but stayed just on/offshore and so the interaction time was taken as the first interaction with the body of the storm.

6) **Lines 293-295: Do the authors have any insights on why warmer storms remain symmetric for longer? The parameter B, determining the onset of ET, basically represents the difference in atmospheric thickness between the left and right sides of the storm, and since all the experiments are conducted under uniformly warmed or cooled conditions, there shouldn't be any temperature gradient differences among them. Additionally, it seems there are no noticeable differences in their locations at the onsets.**

Figure 4 shows that the storms in warmer conditions become stronger than the storms in cooler conditions. Stronger storms can pump more heat into the atmosphere (as seen by the higher values of $-V_T^U$ in Figure 6). This allows them to better condition the atmosphere around them and create a larger environment conducive to strengthening in which the storm can move. Thus the left/right sided difference that is measured for the B parameter takes longer to get close to the core. To better illustrate this, we have added a supplementary figure looking at 600-900 hPa geopotential height differences in circles of 500 km radius for each of the storms at several times (see Figure 6). A scenario-specific time-average of the area +/- 5° around the storm's latitude was subtracted from the geopotential height difference to account for the thermal expansion due to the applied ΔT.

The environmental conditioning is especially visible in the plots of 15 October. The left side of the storm (in relation to storm direction) has a higher proportion of blue (geopotentially thinner, colder air) in the colder scenarios than in the warmer scenarios. We have added a section of this discussion to the manuscript, see lines 329-335.

7) **Lines 345-346: I think this statement is true depending on cases. Typically, TCs undergoing extratropical transition experience an expansion of their wind field, with the radius of maximum winds increasing and the overall wind structure becoming broader and more asymmetric. Additionally, as these storms interact with upper-level waves, their translational speed often accelerates significantly, similar to that of extratropical cyclones. Please consider revising or rephrasing this discussion.**

We thank the reviewer for their contribution to this discussion. We have edited the discussion and it can be found in lines 389-393:

*"TCs bring a different structure and impact footprint than ETCs: in general, TCs are stronger (in terms of wind speed) storms than ETCs. TCs also have only slight wind field asymmetry due to the influence of lower translation speed and vertical wind shear. These influences, however, are larger in ETCs which therefore show larger wind field asymmetries (Jones et al., 2003).*
*PTCs can display a mixture of TC and ETC characteristics, especially while they are still transitioning from one to the other. As they undergo ETT, the radius of maximum wind increases and the entire wind field expands and becomes more asymmetric (Evans et al., 2017). Their translation speed also often increases as a result of interaction with upper level jets, which adds to the wind field to produce stronger wind speeds even as the pressure-induced wind field weakens (Hart & Evans, 2001). As such, PTCs can bring high wind speeds similar to TCs over a large area like ETCs, increasing the potential damages."*

8) **Figure A1: Do the five different initialization times lead to significant differences in storm size? For the simulations initialized on 14 and 15 Oct, no significant differences are seen in tracks. As discussed in the main manuscript, all the storms in these sets of simulations have similar storm size, so they are less affected by the beta drift? If the warmer storms become larger, does the hypothesis that beta drift drives the westward shift of the warmer storms still hold? Beta drift should become more pronounced at higher latitudes.**

We thank the reviewer for their thoughts on this. We have produced several figures to help answer this, these can be found in Appendix A.

Storm size
We investigated this phenomenon by plotting the storm size as in Figure 4(c) for each of the initialization times and for each of the ΔT levels (see Figures R1 and R2).

All initialization times show a difference in storm size, though to varying degrees (Figure R1). This is most pronounced in the earlier initialization times, and only very minimal in the 15 October initialized simulation. We also see that for the same level of ΔT, storms initialized earlier have a larger storm size than those initialized later (Figure R2). This is likely related to the same issue mentioned in lines 460-461 and 471-472 : the earlier the simulation is started, the more time it has to adjust to the alternate climate conditions. Additionally, the earlier the simulation starts, the more time the storms have to diverge from one another.

Beta Drift

While the storm sizes *do* increase with increasing ΔT, beta drift is calculated with the $R_{max}$ instead of the 17 m/s wind contour. Our choice for still using storm size for much of our analysis is motivated by the impacts section of our research: areas outside the $R_{max}$ zone can still experience powerful impacts.

The $R_{max}$ values tend to decrease at the beginning of the simulation, with the warmer simulations showing a much sharper decrease than the cooler simulations (Figure R3, left column). This can also be seen in the contraction of the eye in the storm size plots of Figure R1. At the same time we see an increase in maximum windspeed (Figure R3, middle column). All of these point to a strengthening of the storms due to a conservation of angular momentum. Beta drift, as a combination of these two factors, shows an initial decrease followed by a gradual increase and general stratification by applied ΔT (Figure R3, right column).

In the 14 and 15 October simulations, the difference in $R_{max}$ between the simulations is not as pronounced. Additionally, there is not as much time for the slow expansion of the $R_{max}$ after the initial strengthening decrease, but before the rapid increase also visible in the storm size plots (Figure R1).
Therefore, while we do see an overall larger starting value of the beta drift in the later simulations due to higher latitude and higher starting Vmax, due to the lack of adjustment time we do not see a large spread of track locations as we see in earlier simulations. We have added part of this discussion to the manuscript in Appendix A, see lines 520-535.

9) **Beta drift: It appears that the calculated beta drift among the simulations converges to values within 0.3 m/s after 15th Oct., while the jet streams on 16th show a more diverse distribution (Fig. B2). Could this variation in jet stream distribution be driving the track divergence, rather than the beta drift? Related to this question, Lines 253-256: However, during this period, the warmer storms do not show a noticeable westward shift in their tracks. The significant divergence becomes apparent after the 15th, when the beta drift converges.**

We agree with the reviewer that the jet variation is involved in the track divergence, as explained also in Section 4.2.3. However we believe that the track divergence is due to a combination of both the beta drift and the jet variation.

Figure B3 only included the beta drift and $R_{max}$ until the 16th of October because after this, due to the rapid expansion of the $R_{max}$ associated with the ETT, it was difficult to see the more subtle variations in the beta drift in the tropical and transition phases of the storm. Figure R4 shows the more complete version of Figure B3. We see there that the beta drift values diverge, with the +4 °C simulation obtaining the highest value of peak beta drift, and the -2°C the lowest.

With the exception of the -1°C simulation, the relationship of storms in warmer simulations experiencing stronger beta drift is maintained (see Table 2). The high beta drift of the -1°C scenario can be attributed to an anomalously high $R_{max}$.

**Minor specific points:**

1) **Line 80: Please complete the sentence.**

We completed the sentence by filling in the reference that mistakenly was not inserted. Additionally, we added the equation for the wind speed conversion to make this clearer.

2) **Lines 106-109: Consider rephrasing these lines.**

These lines have been adjusted (see lines 120 – 128):

 *"The initialization time for the RACMO simulations is 12 October 2017 00 UTC. We initially ran six RACMO simulations of Ophelia with initialization times of 00 UTC  on 09 – 14 October 2017. The tracks and central pressure profiles of these simulations are shown in Figure B1.*

*The simulations initialized on 9, 10, and 11 October were not able to capture the observed strengthening of Ophelia as a hurricane and had quite large track deviations from the IBTrACS best track. The simulation initialized on 13 October strengthened rapidly, surpassing the observed pressure values substantially, but also showing large track deviations. The 14 October simulation had a more reasonable track and smaller pressure deficit, but due to its late start it would not be possible to examine the ETT properly. As such, we chose the 12 October initialization. This is quite close to the 12 October 2017 13 UTC initialization time chosen by Rantanen et al. (2020) when they modeled Ophelia, citing a similar problem with lack of strengthening. "*

3) **Figure B1: The IFS is not represented in this figure. Are the solid lines derived from GFS forcing data? Please clarify the figure caption.**

We thank the reviewer for pointing out this mistake, we have adjusted the figure caption to read:

*"Track (a) and minimum central pressure (b) for Hurricane Ophelia for the current climate downscaled RACMO simulations with GFS boundaries with varying initialization times. The black dashed line is the IBTrACS observations, and the pressure time series derived from the ERA5 reanalysis and the GFS analysis data are included in (b)."*

4) **Figure 3: Please add the storm's daily locations to the inset figure.**

We have added the daily locations to the inset figure, both for the RACMO and GFS simulations.

5) **Line 204: This study does not use the PGW approach. Need to be rephrased.**

Following the reviewer's suggestion, we have decided to omit the term PGW and change this to "alternate climate scenarios".

6) **Figure 4(c): Please clarify this figure. It seems the authors are trying to show storm size based on the 17 m/s threshold. What exactly does the y-axis value represent? Is it latitudinal distance, and if so, relative to what? The storm centers?**

We apologize for the confusion. Figure 4(c) shows the extent of 10 minute sustained 17 m/s wind in a slice through the storm centre, in degrees relative to that centre point. As both reviewers raised this concern, the caption has been adjusted to read:

*"Track (a), minimum central pressure (b), north-south diametric slice of extent of 10-minute sustained 17 m/s wind, in degrees relative to storm centre (c), and 10-minute sustained 10m maximum wind (d) of Hurricane Ophelia for the alternate climate downscaled RACMO simulations with GFS boundaries, initialized at 12 October 2017 00 UTC. Dashed line (black dots in (d)) are IBTrACS observations. (a) Circles plotted at 00 UTC of the date indicated. Shading in (c) is the extent of the 0°C simulation."*

7) **Line 249: Please consider providing the formulation applied in this study so that readers can better understand the environmental factors contributing to the shift in the simulated storms.**
We thank the reviewer for the suggestion, and have added the respective formulas to the manuscript

$$BD = 0.72B^{-0.54}r_{max}^2\beta$$

$$\text{where} \quad B = \frac{r_{max}^2\beta}{V_{max}}$$

**8) Lines 345-346 and others: What does Bft mean?**

We apologize for the confusion. Bft is short for Beaufort, a wind categorization system commonly used in (Western) Europe. We will clarify this term in the text, see Line 396)

**Appendix**

[Figure]

Figure R1: North-south diametric slice of extent of 10m instantaneous 17 m/s wind, in degrees relative to storm centre for each of the 5 simulation times and 7 ΔT levels, grouped by simulation time.

[Figure]

Figure R2: As in Figure R1 but grouped by ΔT level.

[Figure]

Figure R3: $R_{max}$, $V_{max}$, and Beta drift for the 5 simulation dates (11-15 October) and 7 ΔT levels (-2 to +4). 6 hour convolution applied to all variables, so plotting starts at 6 hours past initialization time as labelled. Time axis capped at 2017-10-16 to examine tropical phase.

[Figure]

Figure R4: As in Figure R3 but for the full simulation. 6 hour convolution means plotting stops at 16-10-2017 18 UTC.

[Figure]

Figure R5: instantaneous 10m windspeed maps for Ophelia at the time of first contact with land in each of the alternate climate scenarios, with minimum wind speed of 17 m/s. Bottom right is a plot of the times at which the snapshots are taken.

[Figure]

Figure R6: instantaneous 10m windspeed maps for Ophelia at 16 October 00 UTC in each of the alternate climate scenarios, with minimum wind speed of 17 m/s.

[Figure]

Figure R7: Maximum horizontal and vertical pressure gradient values around Ophelia for each simulation. A 6-hour convolution was applied.

Anjana, U., & Kumar, K. K. (2023). New insights into the poleward migration of tropical cyclones and its association with Hadley circulation. *Scientific Reports*, *13*(1), 15009. https://doi.org/10.1038/s41598-023-42323-7

Chan, K. T. F. (2019). Are global tropical cyclones moving slower in a warming climate? *Environmental Research Letters*, *14*(10), 104015. https://doi.org/10.1088/1748-9326/ab4031

Chauvin, F., Pilon, R., Palany, P., & Belmadani, A. (2020). Future changes in Atlantic hurricanes with the rotated-stretched ARPEGE-Climat at very high resolution. *Climate Dynamics*, *54*(1), 947–972. https://doi.org/10.1007/s00382-019-05040-4

Dekker, M. M., Haarsma, R. J., Vries, H. de, Baatsen, M., & Delden, A. J. van. (2018). Characteristics and development of European cyclones with tropical origin in reanalysis data. *Climate Dynamics*, *50*(1), 445–455. https://doi.org/10.1007/s00382-017-3619-8

Evans, C., Wood, K. M., Aberson, S. D., Archambault, H. M., Milrad, S. M., Bosart, L. F., Corbosiero, K. L., Davis, C. A., Pinto, J. R. D., Doyle, J., Fogarty, C., Galarneau, T. J., Grams, C. M., Griffin, K. S., Gyakum, J., Hart, R. E., Kitabatake, N., Lentink, H. S., McTaggart-Cowan, R., … Zhang, F. (2017). *The Extratropical Transition of Tropical Cyclones. Part I: Cyclone Evolution and Direct Impacts*. https://doi.org/10.1175/MWR-D-17-0027.1

Harper, B. A., Kepert, J. D., & Ginger, J. D. (2010). *Guidelines for converting between various wind averaging periods in tropical cyclone conditions* (No. WMO/TD-No. 1555). World Meteorological Organization. https://www.nhc.noaa.gov/verification/verify2.shtml?

Hart, R. E. (2003). A Cyclone Phase Space Derived from Thermal Wind and Thermal

Asymmetry. *Monthly Weather Review*, *131*(4), 585–616.

https://doi.org/10.1175/1520-0493(2003)131<0585:ACPSDF>2.0.CO;2

Hart, R. E., & Evans, J. L. (2001). A Climatology of the Extratropical Transition of Atlantic

Tropical Cyclones. *Journal of Climate*, *14*(4), 546–564.

https://doi.org/10.1175/1520-0442(2001)014<0546:ACOTET>2.0.CO;2

Hart, R. E., Evans, J. L., & Evans, C. (2006). Synoptic Composites of the Extratropical

Transition Life Cycle of North Atlantic Tropical Cyclones: Factors Determining

Posttransition Evolution. *Monthly Weather Review*, *134*(2), 553–578.

https://doi.org/10.1175/MWR3082.1

Kitabatake, N. (2008). Extratropical Transition of Tropical Cyclones in the Western North

Pacific: Their Frontal Evolution. *Monthly Weather Review*, *136*(6), 2066–2090.

https://doi.org/10.1175/2007MWR1958.1

Kossin, J. P. (2018). A global slowdown of tropical-cyclone translation speed. *Nature*,

*558*(7708), 104–107. https://doi.org/10.1038/s41586-018-0158-3

Lanzante, J. R. (2019). Uncertainties in tropical-cyclone translation speed. *Nature*,

*570*(7759), E6–E15. https://doi.org/10.1038/s41586-019-1223-2

Moon, I.-J., Kim, S.-H., & Chan, J. C. L. (2019). Climate change and tropical cyclone

trend. *Nature*, *570*(7759), E3–E5. https://doi.org/10.1038/s41586-019-1222-3

Ritchie, E. A., & Elsberry, R. L. (2007). Simulations of the Extratropical Transition of

Tropical Cyclones: Phasing between the Upper-Level Trough and Tropical

Cyclones. *Monthly Weather Review*, *135*(3), 862–876.

https://doi.org/10.1175/MWR3303.1

Rivière, G. (2011). A Dynamical Interpretation of the Poleward Shift of the Jet Streams in Global Warming Scenarios. *Journal of the Atmospheric Sciences*, *68*(6), 1253–1272. https://doi.org/10.1175/2011JAS3641.1

Sarro, G., & Evans, C. (2022). *An Updated Investigation of Post-Transformation Intensity, Structural, and Duration Extremes for Extratropically Transitioning North Atlantic Tropical Cyclones*. https://doi.org/10.1175/MWR-D-22-0088.1

Stewart, S. R. (2018). *Tropical Cyclone Report: Hurricane Ophelia, 9–15 October 2017*. National Hurricane Center.

Studholme, J., Fedorov, A. V., Gulev, S. K., Emanuel, K., & Hodges, K. (2022). Poleward expansion of tropical cyclone latitudes in warming climates. *Nature Geoscience*, *15*(1), 14–28. https://doi.org/10.1038/s41561-021-00859-1

Sun, Y., Zhong, Z., Li, T., Yi, L., & Shen, Y. (2021). The Slowdown Tends to Be Greater for Stronger Tropical Cyclones. *Journal of Climate*, *34*(14), 5741–5751. https://doi.org/10.1175/JCLI-D-20-0449.1

Tamarin-Brodsky, T., & Kaspi, Y. (2017). Enhanced poleward propagation of storms under climate change. *Nature Geoscience*, *10*(12), 908–913. https://doi.org/10.1038/s41561-017-0001-8

Woollings, T., Drouard, M., O'Reilly, C. H., Sexton, D. M. H., & McSweeney, C. (2023). Trends in the atmospheric jet streams are emerging in observations and could be linked to tropical warming. *Communications Earth & Environment*, *4*(1), 125. https://doi.org/10.1038/s43247-023-00792-8

Yamaguchi, M., Chan, J. C. L., Moon, I.-J., Yoshida, K., & Mizuta, R. (2020). Global warming changes tropical cyclone translation speed. *Nature Communications*, *11*(1), 47. https://doi.org/10.1038/s41467-019-13902-y

Yin, J. H. (2005). A consistent poleward shift of the storm tracks in simulations of 21st century climate. *Geophysical Research Letters*, *32*(18). https://doi.org/10.1029/2005GL023684

Zhang, D., Zhang, H., Zheng, J., Cheng, X., Tian, D., & Chen, D. (2020). Changes in Tropical-Cyclone Translation Speed over the Western North Pacific. *Atmosphere*, *11*(1), Article 1. https://doi.org/10.3390/atmos11010093

Zhang, G., Murakami, H., Knutson, T. R., Mizuta, R., & Yoshida, K. (2020). Tropical cyclone motion in a changing climate. *Science Advances*, *6*(17), eaaz7610. https://doi.org/10.1126/sciadv.aaz7610

---

## Author Response (AR2)

**Reviewer 1**

**Review of Ribberink et al. (2025)**

I would like to thank the authors for performing a thorough revision of the manuscript and replying in detail to the many points raised in the previous review. The paper has made good progress, and it is now closer to being ready for publication.

Despite these improvements, there are still several points that should be addressed before publication.

**1) Use of the term "minimum MSLP": While this is technically not incorrect, it can be redundant or potentially misleading. Since the authors are referring to the storm center, which implies a single-point minimum, the term "mean" is unnecessary. Simply using "minimum sea level pressure (MSLP)" or "central pressure" would be clearer and more appropriate in this context.**

We thank the reviewer for their input and have changed the wording to "central pressure" to match the wording already present in many figure captions.

**2) Lines 235-243: Although the physical mechanisms behind this behavior warrant further investigation (and may be beyond the scope of this manuscript), the significant drop in MSLP does not correspond to a proportional increase in wind speed. I still don't believe the resolution is the whole story. Given that surface wind speed is largely governed by both the pressure gradient and surface properties, one possible explanation is the representation of surface roughness length over the ocean. Assuming RACMO uses the Charnock formulation (as is common in many atmospheric models), the roughness length is tied solely to wind speed. This can lead to unrealistically large roughness values as the storm intensifies. A recent study (Jung et al., 2025) showed that coupling an atmospheric model with an ocean wave model can yield more reasonable surface roughness estimates—specifically, reduced roughness values (see their Figs. 11e–f)—which significantly impacted hurricane wind speed and structure. From this perspective, the lack of air–sea interaction in the current setup may provide a more plausible explanation for the observed wind–pressure inconsistency.**

RACMO does use the Charnock formulation, however after a comparison study found that RACMO underestimated 10m wind speeds at sea, the Charnock formulation was edited slightly to have a constant surface roughness length at higher wind speed values

(van Meijgaard et al., 2008). This was found to increase the 10m wind speeds by about 5-10 % for wind speeds above 20 m/s.

We definitely acknowledge that coupling an ocean model to our atmosphere model would give more accurate results than the slab ocean incorporated in RACMO currently. This applies for more than just windspeed; when Ophelia first forms it moves little, which would likely generate a "cold pool" under the storm, hampering some of its growth. It would be an interesting extension to see how such an edge case storm develops with this feedback, and what that might mean for other similar storms studied in other atmosphere-only models.

**3) Regarding the response to Comment 6):**
**Thanks for the explanation. I understand that stronger storms in warmer conditions can modify their environment more effectively. However, since the warming is applied uniformly across the domain, the environmental temperature gradient (and thus the large-scale thickness asymmetry) remains the same.**
**From my understanding, the authors suggest that internal storm processes can reshape the surrounding geopotential structure in such a way that it delays the development of asymmetries captured by the B parameter. Could you clarify whether there is physical finding that storm-induced heating asymmetrically alters the geopotential field at a sufficient scale to influence the timing of B onset?**

We apologize for the confusion. We were referring to the fact that in strong, mature warm-core systems, the storm creates a larger and more robust warm centre. This is more resistant to the effects of the jet stream, such as wind shear, which can tear into the vertically stacked, axially symmetric core and initiate the thermal asymmetry development. As such it is less that the storm asymmetrically alters the geopotential field than it insulates itself from the factors that would alter the field around itself.

**4) Regarding the response to Comment 8):**
**The response emphasizes that beta drift is calculated using Rmax and largely discusses variations in storm size across experiments. However, beta drift is also sensitive to storm intensity and overall circulation strength. Based on Figure A2, storms in warmer conditions (e.g., October 14) appear to exhibit noticeably greater intensity. In that case, could the authors clarify whether beta drift in these simulations is primarily driven by storm size, or if storm intensity also plays a significant role?**

To demonstrate the relative importance of $r_{max}$ and $V_{max}$ in the beta drift, we use the beta drift equation given as

$$BD = 0.72B^{-0.54}r_{max}^2\beta$$

$$\text{where} \quad B = \frac{r_{max}^2\beta}{V_{max}}$$

If we substitute and simplify:

$$BD = 0.72\left(\frac{r_{max}^2\beta}{V_{max}}\right)^{-0.54} r_{max}^2\beta$$

$$BD = 0.72\left(\frac{V_{max}}{r_{max}^2\beta}\right)^{0.54} r_{max}^2\beta$$

$$BD = 0.72(V_{max})^{0.54}(r_{max}^2\beta)^{0.46}$$

$$BD = 0.72\, V_{max}^{0.54}\, r_{max}^{0.92}\, \beta^{0.46}$$

Due to the greater power on $r_{max}$, we expect that storm size has a greater effect on Beta drift than $V_{max}$.

We have added this to the Appendix (see B3), and a reference in Appendix A, lines 543-545.

*However, based on the beta drift equation, we expect that $R_{max}$ has a greater effect on the beta drift than $V_{max}$ which may be a contributing factor to the relative lack of difference between beta drift values across simulations despite large values of $V_{max}$ (see Appendix B3 for the derivation).*

**Reviewer 2**

**Thank you to the authors for their work addressing comments/concerns raised by myself and another reviewer. I have just one lingering question and one suggestion to address prior to publication.**

**L118–119: Can you comment on this choice to keep CO2 constant for the warmed simulations (perhaps RAMCO doesn't allow for changing concentrations?) and how it might affect your results? I think this information is important to add to the text.**

RACMO does allow for changing concentrations (as was done in e.g. Dullaart et al. (2024)). We use the simple delta T to simplify and constrain the experiment, ensuring that Ophelia is picked up by the jet stream. Adjusting the $CO_2$ levels as well as the delta T would double up and create unrealistic results, and make it difficult to extricate the relative effects of the change in temperature. Changing just $CO_2$ levels would be similar to a PGW approach.

**Fig. 4 and related discussion: Michaelis et al. (2019; https://gmd.copernicus.org/articles/12/3725/2019/) and included references might be useful for other examples of misrepresenting TC intensity (especially regarding max winds).**

We thank you for the suggestion and have inserted references into that section, specifically lines 245-247.

Dullaart, J. C. M., de Vries, H., Bloemendaal, N., Aerts, J. C. J. H., & Muis, S. (2024). Improving our understanding of future tropical cyclone intensities in the Caribbean using a high-resolution regional climate model. *Scientific Reports*, *14*(1), 6108. https://doi.org/10.1038/s41598-023-49685-y

van Meijgaard, E., van Ulft, L. H., van de Berg, W. J., Bosveld, F. C., van den Hurk, B. J. J. M., Lenderink, G., & Siebesma, A. P. (2008). *The KNMI regional atmospheric climate model RACMO version 2.* (No. TR-302).